# Learning to Control PDEs
# with Differentiable Physics

**Philipp Holl**
Technical University of Munich

**Vladlen Koltun**
Intel Labs

**Nils Thuerey**
Technical University of Munich

## Abstract

Predicting outcomes and planning interactions with the physical world are long-standing goals for machine learning. A variety of such tasks involves continuous physical systems, which can be described by partial differential equations (PDEs) with many degrees of freedom. Existing methods that aim to control the dynamics of such systems are typically limited to relatively short time frames or a small number of interaction parameters. We present a novel hierarchical predictor-corrector scheme which enables neural networks to learn to understand and control complex nonlinear physical systems over long time frames. We propose to split the problem into two distinct tasks: planning and control. To this end, we introduce a predictor network that plans optimal trajectories and a control network that infers the corresponding control parameters. Both stages are trained end-to-end using a differentiable PDE solver. We demonstrate that our method successfully develops an understanding of complex physical systems and learns to control them for tasks involving PDEs such as the incompressible Navier-Stokes equations.

## 1 Introduction

Intelligent systems that operate in the physical world must be able to perceive, predict, and interact with physical phenomena (Battaglia et al., 2013). In this work, we consider physical systems that can be characterized by partial differential equations (PDEs). PDEs constitute the most fundamental description of evolving systems and are used to describe every physical theory, from quantum mechanics and general relativity to turbulent flows (Courant & Hilbert, 1962; Smith, 1985). We aim to endow artificial intelligent agents with the ability to direct the evolution of such systems via continuous controls.

Such optimal control problems have typically been addressed via iterative optimization. Differentiable solvers and the adjoint method enable efficient optimization of high-dimensional systems (Toussaint et al., 2018; de Avila Belbute-Peres et al., 2018; Schenck & Fox, 2018). However, direct optimization through gradient descent (single shooting) at test time is resource-intensive and may be difficult to deploy in interactive settings. More advanced methods exist, such as multiple shooting and collocation, but they commonly rely on modeling assumptions that limit their applicability, and still require computationally intensive iterative optimization at test time.

Iterative optimization methods are expensive because they have to start optimizing from scratch and typically require a large number of iterations to reach an optimum. In many real-world control problems, however, agents have to repeatedly make decisions in specialized environments, and reaction times are limited to a fraction of a second. This motivates the use of data-driven models such as deep neural networks, which combine short inference times with the capacity to build an internal representation of the environment.

We present a novel deep learning approach that can learn to represent solution manifolds for a given physical environment, and is orders of magnitude faster than iterative optimization techniques. The core of our method is a hierarchical predictor-corrector scheme that temporally divides the problem into easier subproblems. This enables us to combine models specialized to different time scales in order to control long sequences of complex high-dimensional systems. We train our models using a differentiable PDE solver that can provide the agent with feedback of how interactions at any point in time affect the outcome. Our models learn to represent manifolds containing a large number of solutions, and can thereby avoid local minima that can trap classic optimization techniques.

We evaluate our method on a variety of control tasks in systems governed by advection-diffusion PDEs such as the Navier-Stokes equations. We quantitatively evaluate the resulting sequences on how well they approximate the target state and how much force was exerted on the physical system. Our method yields stable control for significantly longer time spans than alternative approaches.

## 2   BACKGROUND

Physical problems commonly involve nonlinear PDEs, often with many degrees of freedom. In this context, several works have proposed methods for improving the solution of PDE problems (Long et al., 2018; Bar-Sinai et al., 2019; Hsieh et al., 2019) or used PDE formulations for unsupervised optimization (Raissi et al., 2018). Lagrangian fluid simulation has been tackled with regression forests (Ladicky et al., 2015), graph neural networks (Mrowca et al., 2018; Li et al., 2019), and continuous convolutions (Ummenhofer et al., 2020). Data-driven turbulence models were trained with MLPs (Ling et al., 2016). Fully-convolutional networks were trained for pressure inference (Tompson et al., 2017) and advection components were used in adversarial settings (Xie et al., 2018). Temporal updates in reduced spaces were learned via the Koopman operator (Morton et al., 2018). In a related area, deep networks have been used to predict chemical properties and the outcome of chemical reactions (Gilmer et al., 2017; Bradshaw et al., 2019).

Differentiable solvers have been shown to be useful in a variety of settings. Degrave et al. (2019) and de Avila Belbute-Peres et al. (2018) developed differentiable simulators for rigid body mechanics. (See Popovic et al. (2000) for earlier work in computer graphics.) Toussaint et al. (2018) applied related techniques to manipulation planning. Specialized solvers were developed to infer protein structures (Ingraham et al., 2019), interact with liquids (Schenck & Fox, 2018), control soft robots (Hu et al., 2019), and solve inverse problems that involve cloth (Liang et al., 2019). Like ours, these works typically leverage the automatic differentiation of deep learning pipelines (Griewank & Walther, 2008; Maclaurin et al., 2015; Amos & Kolter, 2017; Mensch & Blondel, 2018; van Merriënboer et al., 2018; Chen et al., 2018; Bradbury et al., 2018; Paszke et al., 2019; Tokui et al., 2019). However, while the works above target Lagrangian solvers, i.e. reference frames moving with the simulated material, we address grid-based solvers, which are particularly appropriate for dense, volumetric phenomena.

The adjoint method (Lions, 1971; Pironneau, 1974; Jameson, 1988; Giles & Pierce, 2000; Bewley, 2001; McNamara et al., 2004) is used by most machine learning frameworks, where it is commonly known as reverse mode differentiation (Werbos, 2006; Chen et al., 2018). While a variety of specialized adjoint solvers exist (Griewank et al., 1996; Fournier et al., 2012; Farrell et al., 2013), these packages do not interface with production machine learning frameworks. A supporting contribution of our work is a differentiable PDE solver called $\Phi_{\text{Flow}}$ that integrates with TensorFlow (Abadi et al., 2016) and PyTorch (Paszke et al., 2019). It is publicly available at https://github.com/tumpbs/PhiFlow.

## 3   PROBLEM

Consider a physical system $\boldsymbol{u}(\boldsymbol{x}, t)$ whose natural evolution is described by the PDE

$$\frac{\partial \boldsymbol{u}}{\partial t} = \mathcal{P}\left(\boldsymbol{u}, \frac{\partial \boldsymbol{u}}{\partial \boldsymbol{x}}, \frac{\partial^2 \boldsymbol{u}}{\partial \boldsymbol{x}^2}, ..., \boldsymbol{y}(t)\right), \tag{1}$$

where $\mathcal{P}$ models the physical behavior of the system and $\boldsymbol{y}(t)$ denotes external factors that can influence the system. We now introduce an agent that can interact with the system by controlling certain parameters of the dynamics. This could be the rotation of a motor or fine-grained control over a field. We factor out this influence into a force term $\boldsymbol{F}$, yielding

$$\frac{\partial \boldsymbol{u}}{\partial t} = \mathcal{P}\left(\boldsymbol{u}, \frac{\partial \boldsymbol{u}}{\partial \boldsymbol{x}}, \frac{\partial^2 \boldsymbol{u}}{\partial \boldsymbol{x}^2}, ...\right) + \boldsymbol{F}(t). \tag{2}$$

The agent can now be modelled as a function that computes $\boldsymbol{F}(t)$. As solutions of nonlinear PDEs were shown to yield low-dimensional manifolds (Foias et al., 1988; Titi, 1990), we target solution manifolds of $\boldsymbol{F}(t)$ for a given choice of $\mathcal{P}$ with suitable boundary conditions. This motivates our choice to employ deep networks for our agents.

In most real-world scenarios, it is not possible to observe the full state of a physical system. When considering a cloud of smoke, for example, the smoke density may be observable while the velocity field may not be seen directly. We model the imperfect information by defining the observable state of $\boldsymbol{u}$ as $\boldsymbol{o}(\boldsymbol{u})$. The observable state is problem dependent, and our agent is conditioned only on these observations, i.e. it does not have access to the full state $\boldsymbol{u}$.

Using the above notation, we define the control task as follows. An initial observable state $\boldsymbol{o}_0$ of the PDE as well as a target state $\boldsymbol{o}^*$ are given (Figure 1a). We are interested in a reconstructed trajectory $\boldsymbol{u}(t)$ that matches these states at $t_0$ and $t_*$, i.e. $\boldsymbol{o}_0 = \boldsymbol{o}(\boldsymbol{u}(t_0)), \boldsymbol{o}^* = \boldsymbol{o}(\boldsymbol{u}(t_*))$, and minimizes the amount of force applied within the simulation domain $\mathcal{D}$ (Figure 1b):

$$L_{\boldsymbol{F}}[\boldsymbol{u}(t)] = \int_{t_0}^{t_*} \int_{\mathcal{D}} |\boldsymbol{F}_{\boldsymbol{u}}(t)|^2 \, dx \, dt. \tag{3}$$

Taking discrete time steps $\Delta t$, the reconstructed trajectory $\boldsymbol{u}$ is a sequence of $n = (t_* - t_0)/\Delta t$ states.

When an observable dimension cannot be controlled directly, there may not exist any trajectory $\boldsymbol{u}(t)$ that matches both $\boldsymbol{o}_0$ and $\boldsymbol{o}^*$. This can stem from either physical constraints or numerical limitations. In these cases, we settle for an approximation of $\boldsymbol{o}^*$. To measure the quality of the approximation of the target, we define an observation loss $L_{\boldsymbol{o}}^*$. The form of this loss can be chosen to fit the problem. We combine Eq. 3 and the observation loss into the objective function

$$L[\boldsymbol{u}(t)] = \alpha \cdot L_{\boldsymbol{F}}[\boldsymbol{u}(t)] + L_{\boldsymbol{o}}^*(\boldsymbol{u}(t_*)), \quad (4)$$

Figure 1: Illustration of possible trajectories. The grey lines represent the unperturbed evolution of the physical system. The amount of applied force corresponds to how far the trajectory deviates from the natural evolution.

with $\alpha > 0$. We use square brackets to denote functionals, i.e. functions depending on fields or series rather than single values.

## 4 PRELIMINARIES

**Differentiable solvers.** Let $\boldsymbol{u}(\boldsymbol{x}, t)$ be described by a PDE as in Eq. 1. A regular solver can move the system forward in time via Euler steps:

$$\boldsymbol{u}(t_{i+1}) = \text{Solver}[\boldsymbol{u}(t_i), \boldsymbol{y}(t_i)] = \boldsymbol{u}(t_i) + \Delta t \cdot \mathcal{P}(\boldsymbol{u}(t_i), ..., \boldsymbol{y}(t_i)). \tag{5}$$

Each step moves the system forward by a time increment $\Delta t$. Repeated execution produces a trajectory $u(t)$ that approximates a solution to the PDE. This functionality for time advancement by itself is not well-suited to solve optimization problems, since gradients can only be approximated by finite differencing. For high-dimensional or continuous systems, this method becomes computationally expensive because a full trajectory needs to be computed for each optimizable parameter.

Differentiable solvers resolve this issue by solving the adjoint problem (Pontryagin, 1962) via analytic derivatives. The adjoint problem computes the same mathematical expressions while working with lower-dimensional vectors. A differentiable solver can efficiently compute the derivatives with respect to any of its inputs, i.e. $\partial \boldsymbol{u}(t_{i+1})/\partial \boldsymbol{u}(t_i)$ and $\partial \boldsymbol{u}(t_{i+1})/\partial \boldsymbol{y}(t_i)$. This allows for gradient-based optimization of inputs or control parameters over an arbitrary number of time steps.

**Iterative trajectory optimization.** Many techniques exist that try to find optimal trajectories by starting with an initial guess for $\boldsymbol{F}(t)$ and slightly changing it until reaching an optimum. The simplest of these is known as single shooting. In one optimization step, it simulates the full dynamics, then backpropagates the loss through the whole sequence to optimize the controls (Kraft, 1985; Leineweber et al., 2003). Replacing $\boldsymbol{F}(t)$ with an agent $\boldsymbol{F}(t|\boldsymbol{o}_t, o^*)$, which can be parameterized by a deep network, yields a simple training method. For a sequence of $n$ frames, this setup contains $n$ linked copies of the agent and is depicted in Figure 2. We refer to such an agent as a control force estimator (CFE).

Optimizing such a chain of CFEs is both computationally expensive and causes gradients to pass through a potentially long sequence of highly nonlinear simulation steps. When the reconstruction $u$ is close to an optimal trajectory, this is not a problem because the gradients $\Delta u$ are small and the operations executed by the solver are differentiable by construction. The solver can therefore be locally approximated by a first-order polynomial and the gradients can be safely backpropagated. For large $\Delta u$, e.g. at the beginning of an optimization, this approximation breaks down, causing the gradients to become unstable while passing through the chain. This instability in the training process can prevent single-shooting approaches from converging and deep networks from learning unless they are initialized near an optimum.

Alternatives to single shooting exist, promising better and more efficient convergence. Multiple shooting (Bock & Plitt, 1984) splits the trajectory into segments with additional defect constraints. Depending on the physical system, this method may have to be adjusted for specific problems (Treuille et al., 2003). Collocation schemes (Hargraves & Paris, 1987) model trajectories with splines. While this works well for particle trajectories, it is poorly suited for Eulerian solvers where the evolution of individual points does not reflect the overall motion. Model reduction can be used to reduce the dimensionality or nonlinearity of the problem, but generally requires domain-specific knowledge. When applicable, these methods can converge faster or in a more stable manner than single shooting. However, as we are focusing on a general optimization scheme in this work, we will use single shooting and its variants as baseline comparisons.

**Supervised and differentiable physics losses.** One of the key ingredients in training a machine learning model is the choice of loss function. For many tasks, supervised losses are used, i.e. losses that directly compare the output of the model for a specific input with the desired ground truth. While supervised losses can be employed for trajectory optimization, far better loss functions are possible when a differentiable solver is available. We will refer to these as *differentiable physics* loss functions. In this work, we employ a combination of supervised and differentiable physics losses, as both come with advantages and disadvantages.

One key limitation of supervised losses is that they can only measure the error of a single time step. Therefore, an agent cannot get any measure of how its output would influence future time steps. Another problem arises from the form of supervised training data which comprises input-output pairs, which may be obtained directly from data generation or through iterative optimization. Since optimal control problems are generally not unimodal, there can exist multiple possible outputs for one input. This ambiguity in the supervised training process will lead to suboptimal predictions as the network will try to find a compromise between all possible outputs instead of picking one of them.

Differentiable physics losses solve these problems by allowing the agent to be directly optimized for the desired objective (Eq. 4). Unlike supervised losses, differentiable physics losses require a differentiable solver to backpropagate the gradients through the simulation. Multiple time steps can be chained together, which is a key requirement since the objective (Eq. 4) explicitly depends on all time steps through $L_F[u(t)]$ (Eq. 3). As with iterative solvers, one optimization step for a sequence of $n$ frames then invokes the agent $n$ times before computing the loss, each invocation followed by a solver step. The employed differentiable solver backpropagates the gradients through the whole sequence, which gives the model feedback on (i) how its decisions change the future trajectory and

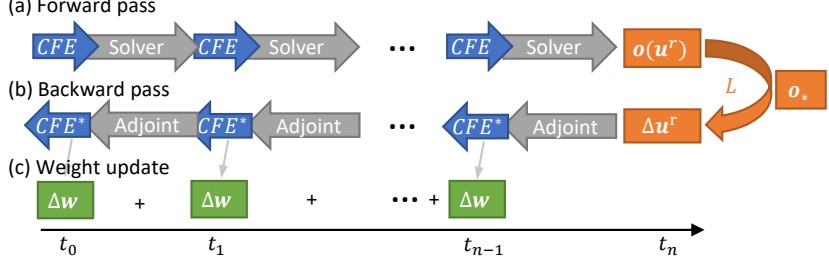

Figure 2: Single-shooting optimization with a control force estimator (CFE). (a) The forward pass simulates the full sequence. (b) Backpropagation computes the adjoint problem. (c) Weight updates are accumulated and applied to the CFE.

(ii) how to handle states as input that were reached because of its previous decisions. Since no ground truth needs to be provided, multi-modal problems naturally converge towards one solution.

## 5  METHOD

In order to optimally interact with a physical system, an agent has to (i) build an internal representation of an optimal observable trajectory $o(u(t))$ and (ii) learn what actions to take to move the system along the desired trajectory. These two steps strongly resemble the predictor-corrector method (Press et al., 2007). Given $o(t)$, a predictor-corrector method computes $o(t + \Delta t)$ in two steps. First, a prediction step approximates the next state, yielding $o^p(t + \Delta t)$. Then, the correction uses $o^p(t + \Delta t)$ to refine the initial approximation and obtain $o(t + \Delta t)$. Each step can, to some degree, be learned independently.

This motivates splitting the agent into two neural networks: an observation predictor (OP) network that infers intermediate states $o^p(t_i)$, $i \in \{1, 2, ...n - 1\}$, planning out a trajectory, and a corrector network (CFE) that estimates the control force $F(t_i|o(u_i), o_{i+1}^p)$ to follow that trajectory as closely as possible. This splitting has the added benefit of exposing the planned trajectory, which would otherwise be inaccessible. As we will demonstrate, it is crucial for the prediction stage to incorporate knowledge about longer time spans. We address this by modelling the prediction as a temporally hierarchical process, recursively dividing the problem into smaller subproblems.

To achieve this, we let the OP not directly infer $o^p(t_{i+1} \,|\, o(u_i), o^*)$ but instead model it to predict the optimal center point between two states at times $t_i, t_j$, with $i, j \in \{1, 2, ...n - 1\}, j > i$, i.e. $o^p((t_i + t_j)/2 \,|\, o_i, o_j)$. This function is much more general than predicting the state of the next time step since two arbitrary states can be passed as arguments. Recursive OP evaluations can then partition the sequence until a prediction $o^p(t_i)$ for every time step $t_i$ has been made.

This scheme naturally enables scaling to arbitrary time frames or arbitrary temporal resolutions, assuming that the OP can correctly anticipate the physical behavior. Since physical systems often exhibit different behaviors on different time scales and the OP can be called with states separated by arbitrary time spans, we condition the OP on the time scale it is evaluated on by instantiating and training a unique version of the model for every time scale. This simplifies training and does not significantly increase the model complexity as we use factors of two for the time scales, and hence the number of required models scales with $\mathcal{O}(\log_2 n)$. We will refer to one instance of an $\text{OP}_n$ by the time span between its input states, measured in the number of frames $n = (t_j - t_i)/\Delta t$.

**Execution order.**   With the CFE and $\text{OP}_n$ as building blocks, many algorithms for solving the control problem, i.e. for computing $F(t)$, can be assembled and trained. We compared a variety of algorithms and found that a scheme we will refer to as *prediction refinement* produces the best results. It is based on the following principles: (i) always use the finest scale OP possible to make a prediction, (ii) execute the CFE followed by a solver step as soon as possible, (iii) refine predictions after the solver has computed the next state. The algorithm that realizes these goals is shown in Appendix B with an example for $n = 8$. To understand the algorithm and resulting execution orders, it is helpful to consider simpler algorithms first.

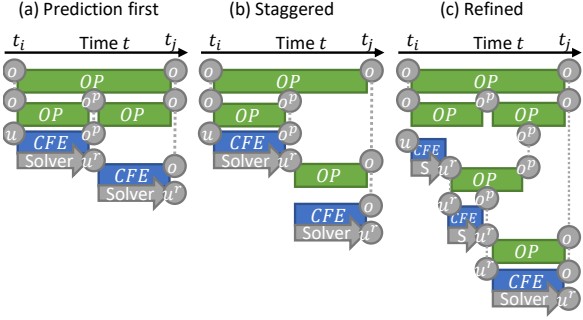

Figure 3: Overview of the different execution schemes.

The simplest combination of CFE and $\text{OP}_n$ invocations that solves the full trajectory, shown in Figure 3a, can be described as follows. Initially, all intermediate states are predicted hierarchically. The first prediction is the half-way point $o^p(t_{n/2} \,|\, o_0, o^*)$, generated by the $\text{OP}_n$. Using that as input to an $\text{OP}_{n/2}$ results in new predictions at $t_{n/4}, t_{3n/4}$. Continuing with this scheme, a prediction can be made for each $t_i, i \in 1, ..., n - 1$. Next, the actual trajectory is evaluated step by step. For each step $t_i$, the CFE computes the control force $F(t_i)$ conditioned on the state at $t_i$ and the prediction $o^p(t_{i+1})$. Once $F(t_i)$ is known, the solver can step the simulation to the next state at $t_{i+1}$. This al-

gorithm finds a trajectory in time $\mathcal{O}(n)$ since $n$ CFE calls and $n-1$ OP calls are required in total (see Appendix B). However, there are inherent problems with this algorithm. The physical constraints of the PDE and potential approximation errors of the CFE can result in observations that are only matched partially. This can result in the reconstructed trajectory exhibiting undesirable oscillations, often visible as jittering. When subsequent predictions do not line up perfectly, large forces may be applied by the CFE or the reconstructed trajectory might stop following the predictions altogether.

This problem can be alleviated by changing the execution order of the two-stage algorithm described above. The resulting algorithm is shown in Figure 3b and will be referred to as *staggered execution*. In this setup, the simulation is advanced as soon as a prediction for the next observable state exists and OPs are only executed when their state at time $t_i$ is available. This staggered execution scheme allows future predictions to take deviations from the predicted trajectory into account, preventing a divergence of the actual evolution $o(u(t))$ from the prediction $o^p(t)$.

While the staggered execution allows most predictions to correct for deviations from the predicted trajectory $o^p$, this scheme leaves several predictions unmodified. Most notably, the prediction $o^p(t_{n/2})$, which is inferred from just the initial state and the desired target, remains unchanged. This prediction must therefore be able to guide the reconstruction in the right direction without knowing about deviations in the system that occurred up to $t_{n/2-1}$. As a practical consequence, a network trained with this scheme typically learns to average over the deviations, resulting in blurred predictions (see Appendix D.2).

---

**Algorithm 1:** Recursive algorithm computing the prediction refinement. The algorithm is called via $\mathrm{Reconstruct}[o_0, o_*, absent]$ to reconstruct a full trajectory from $o_0$ to $o_*$.

---

function $\mathrm{Reconstruct}[o(u_0), o_n, o_{2n}]$;
**Input** : Initial observation $o(u_0)$, observation $o_n$, optional observation $o_{2n}$
**Output:** Observation of the reconstructed state $o(u_n)$
**if** $n = 1$ **then**
    $F \leftarrow \mathrm{CFE}[o(u_0), o_1]$
    $u_1 \leftarrow \mathrm{Solver}[u_0, F]$
    **return** $o(u_1)$
**else**
    $o_{n/2} \leftarrow \mathrm{OP}[o(u_0), o_n]$
    $o(u_{n/2}) \leftarrow \mathrm{Reconstruct}[o(u_0), o_{n/2}, o_n]$
    **if** $o_{2n}$ *present* **then**
        $o_{3n/2} \leftarrow \mathrm{OP}[o_n, o_{2n}]$
        $o_n \leftarrow \mathrm{OP}[o(u_{n/2}), o_{3n/2}]$
    **else**
        $o_{3n/2} \leftarrow absent$
    **end**
    $o(u_n) \leftarrow \mathrm{Reconstruct}[o(u_{n/2}), o_n, o_{3n/2}]$
    **return** $o(u_n)$
**end**

---

The prediction refinement scheme, listed in Algorithm 1 and illustrated in Figure 3c, solves this problem by re-evaluating existing predictions whenever the simulation progesses in time. Not all predictions need to be updated, though, and an update to a prediction at a finer time scale can depend on a sequence of other predictions. The *prediction refinement* algorithm that achieves this in an optimal form is listed in Appendix B. While the resulting execution order is difficult to follow for longer sequences with more than $n = 8$ frames, we give an overview of the algorithm by considering the prediction for time $t_{n/2}$. After the first center-frame prediction $o^p(t_{n/2})$ of the $n$-frame sequence is made by $\mathrm{OP}_n$, the prediction refinement algorithm calls itself recursively until all frames up to frame $n/4$ are reconstructed from the CFE and the solver. The center prediction is then updated using $\mathrm{OP}_{n/2}$ for the next smaller time scale compared to the previous prediction. The call of $\mathrm{OP}_{n/2}$ also depends on $o^p(t_{3n/4})$, which was predicted using $\mathrm{OP}_{n/2}$. After half of the remaining distance to the center is reconstructed by the solver, the center prediction at $t_{n/2}$ is updated again, this time by the $\mathrm{OP}_{n/4}$, including all prediction dependencies. Hence, the center prediction is continually refined every time the temporal distance between the latest reconstruction and the prediction halves, until

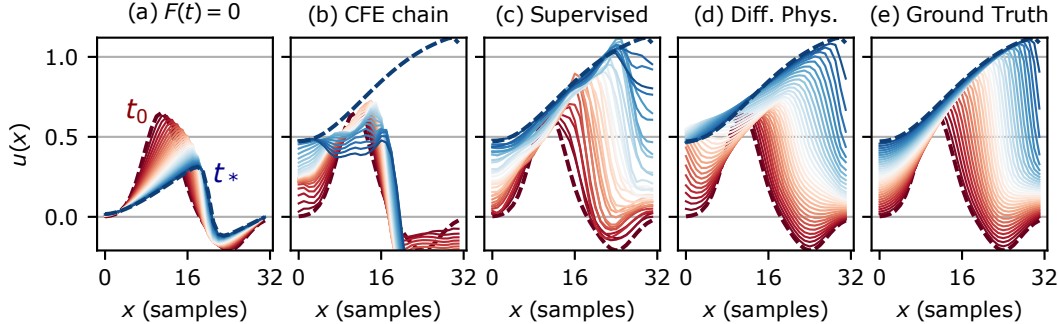

Figure 4: Trajectories for an example control task using Burger's equation. Initial and target states are plotted with thick dashed lines in red and blue, respectively. Inferred states are shown as solid lines. (a) Natural evolution. (b) Reconstruction using a CFE chain. (c,d) Reconstructions using our hierarchical predictor-corrector scheme. (e) Ground-truth trajectory generated with constant force.

the reconstruction reaches that frame. This way, all final predictions $o^p(t_i)$ are conditioned on the reconstruction of the previous state $u(t_{i-1})$ and can therefore account for all previous deviations.

The prediction refinement scheme requires the same number of force inferences but an increased number of OP evaluations compared to the simpler algorithms. With a total of $3n - 2\log_2(n) - 3$ OP evaluations (see Appendix B), it is of the same complexity, $\mathcal{O}(n)$. In practice, this refinement scheme incurs only a small overhead in terms of computation, which is outweighed by the significant gains in quality of the learned control function.

## 6 RESULTS

We evaluate the capabilities of our method to learn to control physical PDEs in three different test environments of increasing complexity. We first target a simple but nonlinear 1D equation, for which we present an ablation study to quantify accuracy. We then study two-dimensional problems: an incompressible fluid and a fluid with complex boundaries and indirect control. Full details are given in Appendix D. Supplemental material containing additional sequences for all of the tests can be downloaded from https://ge.in.tum.de/publications/2020-iclr-holl.

**Burger's equation.** Burger's equation is a nonlinear PDE that describes the time evolution of a single field, $u$ (LeVeque, 1992). Using Eq. 1, it can be written as

$$\mathcal{P}\left(u, \frac{\partial u}{\partial x}, \frac{\partial^2 u}{\partial x^2}\right) = -u \cdot \frac{\partial u}{\partial x} + \nu \frac{\partial^2 u}{\partial x^2}. \tag{6}$$

Examples of the unperturbed evolution are shown in Figure 4a. We let the whole state be observable and controllable, i.e. $o(t) = u(t)$, which implies that $o^*$ can always be reached exactly.

The results of our ablation study with this equation are shown in Table 1. The table compares the resulting forces applied by differently trained models when reconstructing a ground-truth sequence (Figure 4e). The variant denoted by *CFE chain* uses a neural network to infer the force without any intermediate predictions. With a supervised loss, this method learns to approximate a single step well. However, for longer sequences, results quickly deviate from an ideal trajectory and diverge because the network never learned to account for errors made in previous steps (Figure 4b). Training the network with the objective loss (Eq. 4) using the differentiable solver greatly increases the quality of the reconstructions. On average, it applies only 34% of the force used by the supervised model as it learns to correct the temporal evolution of the PDE model.

Next, we evaluate variants of our predictor-corrector approach, which hierarchically predicts intermediate states. Here, the CFE is implemented as $F(t_i) = (o^p(t_{i+1}) - u(t_i))/\Delta t$. Unlike the simple CFE chain above, training with the supervised loss and staggered execution produces stable (albeit jittering) trajectories that successfully converge to the target state (Figure 4c). Surprisingly, this supervised method reaches almost the same accuracy as the differentiable CFE, despite not having access to physics-based gradients. However, employing the differentiable physics loss greatly

Table 1: Quantitative reconstruction evaluation using Burger's equation, avg. for 100 examples.

| Execution scheme | Training loss | Force $\int |\boldsymbol{F}| \, dt$ | Inference time (ms) |
|---|---|---|---|
| CFE chain | Supervised | $83.4 \pm 2.0$ | $0.024 \pm 0.013$ |
| CFE chain | Diff. Physics | $28.8 \pm 0.8$ | $0.024 \pm 0.013$ |
| Staggered | Supervised | $34.3 \pm 1.1$ | $1.15 \pm 0.19$ |
| Staggered | Diff. Physics | $15.3 \pm 0.7$ | $1.15 \pm 0.19$ |
| Refined | Diff. Physics | $14.2 \pm 0.7$ | $3.05 \pm 0.37$ |
| Iterative optim. (60 iter.) | Diff. Physics | $15.3 \pm 1.6$ | $52.7 \pm 2.1$ |
| Iterative optim. (300 iter.) | Diff. Physics | $10.2 \pm 1.9$ | $264.0 \pm 3.0$ |

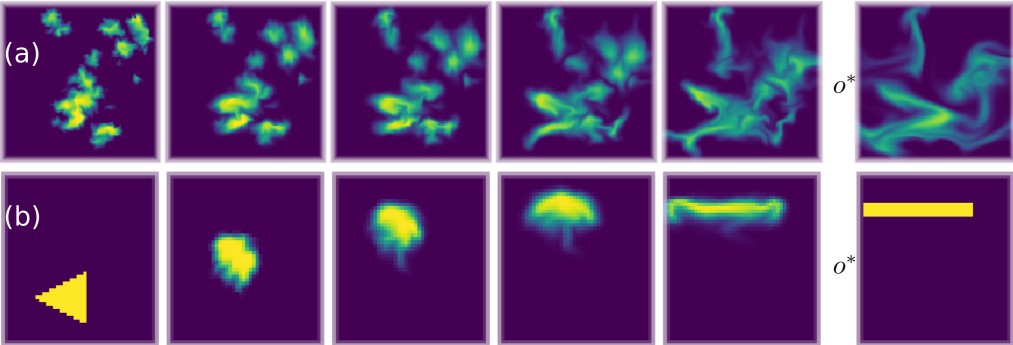

Figure 5: Example reconstructed trajectory from (a) the natural flow test set and (b) the shape test set. The target state $\boldsymbol{o}^*$ is shown on the right.

improves the reconstruction quality, producing solutions that are hard to distinguish from ideal trajectories (Figure 4d). The prediction refinement scheme further improves the accuracy, but the differences to the staggered execution are relatively small as the predictions of the latter are already very accurate.

Table 1 also lists the results of classic shooting-based optimization applied to this problem. To match the quality of the staggered execution scheme, the shooting method requires around 60 optimization steps. These steps are significantly more expensive to compute, despite the fast convergence. After around 300 iterations, the classic optimization reaches an optimal value of 10.2 and the loss stops decreasing. Starting the iterative optimization with our method as an initial guess pushes the optimum slightly lower to 10.1. Thus, even this relatively simple problem shows the advantages of our learned approach.

**Incompressible fluid flow.** Next, we apply our algorithm to two-dimensional fluid dynamics problems, which are challenging due to the complexities of the governing Navier-Stokes equations (Batchelor, 1967). For a velocity field $\boldsymbol{v}$, these can be written as

$$\mathcal{P}(\boldsymbol{v}, \nabla \boldsymbol{v}) = -\boldsymbol{v} \cdot \nabla \boldsymbol{v} + \nu \nabla^2 \boldsymbol{v} - \nabla p, \tag{7}$$

subject to the hard constraints $\nabla \cdot \boldsymbol{v} = 0$ and $\nabla \times p = 0$, where $p$ denotes pressure and $\nu$ the viscosity. In addition, we consider a passive density $\rho$ that moves with the fluid via $\partial \rho / \partial t = -\boldsymbol{v} \cdot \nabla \rho$. We set $\boldsymbol{v}$ to be hidden and $\rho$ to be observable, and allow forces to be applied to all of $\boldsymbol{v}$.

We run our tests on a $128^2$ grid, resulting in more than 16,000 effective continuous control parameters. We train the OP and CFE networks for two different tasks: reconstruction of natural fluid flows and controlled shape transitions. Example sequences are shown in Figure 5 and a quantitative evaluation, averaged over 100 examples, is given in Table 2. While all methods manage to approximate the target state well, there are considerable differences in the amount of force applied. The supervised technique exerts significantly more force than the methods based on the differentiable solver, resulting in jittering reconstructions. The prediction refinement scheme produces the smoothest transitions, converging to about half the loss of the staggered, non-refined variant.

We compare our method to classic shooting algorithms for this incompressible flow problem. While a direct shooting method fails to converge, a more advanced multi-scale shooting approach still requires 1500 iterations to obtain a level of accuracy that our model achieves almost instantly. In

Table 2: A comparison of methods in terms of final cost for (a) the natural flow setup and (b) the shape transitions. The initial distribution is sampled randomly and evolved to the target state.

| Execution | Loss | a) Force $L_F$ | a) Obs. $L_o^*$ | b) Force $L_F$ | b) Obs. $L_o^*$ |
|---|---|---|---|---|---|
| Staggered | Supervised | $243 \pm 11$ | $1.53 \pm 0.23$ | n/a | n/a |
| Staggered | Diff. Physics | $22.6 \pm 1.1$ | $0.64 \pm 0.08$ | $89 \pm 6$ | $0.331 \pm 0.134$ |
| Refined | Diff. Physics | $11.7 \pm 0.6$ | $0.88 \pm 0.11$ | $75 \pm 4$ | $0.126 \pm 0.010$ |



Figure 6: Indirect control sequence. Obstacles are marked white, control regions light blue. The white arrows indicate the velocity field. The domain is enclosed in a solid box with an open top.

addition, our model successfully learns a solution manifold, while iterative optimization techniques essentially *start from scratch* every time. This global view leads our model to more intuitive solutions and decreases the likelihood of convergence to undesirable local minima. The solutions of our method can also be used as initial guesses for iterative solvers, as illustrated in Appendix D.4. We find that the iterative optimizer with an initial guess converges to solutions that require only 57.4% of the force achieved by the iterative optimizer with default initialization. This illustrates how the more global view of the learned solution manifold can improve the solutions of regular optimization runs.

Splitting the task into prediction and correction ensures that intermediate predicted states are physically plausible and allows us to generalize to new tasks. For example, we can infer transitions involving multiple shapes, despite training only on individual shapes. This is demonstrated in Appendix D.2.

**Incompressible fluid with indirect control.** The next experiment increases the complexity of the fluid control problem by adding obstacles to the simulated domain and limiting the area that can be controlled by the network. An example sequence in this setting is shown in Figure 6. As before, only the density $\rho$ is observable. Here, the goal is to move the smoke from its initial position near the center into one of the three "buckets" at the top. Control forces can only be applied in the peripheral regions, which are outside the visible smoke distribution. Only by synchronizing the 5000 continuous control parameters can a directed velocity field be constructed in the central region.

We first infer trajectories using a trained CFE network and predictions that move the smoke into the desired bucket in a straight line. This baseline manages to transfer $89\% \pm 2.6\%$ of the smoke into the target bucket. Next we enable the hierarchical predictions and train the OPs. This version manages to maneuver $99.22\% \pm 0.15\%$ of the smoke into the desired buckets while requiring $19.1\% \pm 1.0\%$ less force.

For comparison, Table 3 also lists success rate and execution time for a direct optimization. Despite only obtaining a low success rate of 82%, the shooting method requires several orders of magnitude longer than evaluating our trained model. Since all optimizations are independent of each other, some find better solutions than others, reflected in the higher standard deviation. The increased number of free parameters and complexity of the fluid dynamics to be controlled make this problem intractable for the shooting method, while our model can leverage the learned representation to infer a solution very quickly. Further details are given in Appendix D.3.

## 7 CONCLUSIONS

We have demonstrated that deep learning models in conjunction with a differentiable physics solver can successfully predict the behavior of complex physical systems and learn to control them. The in-

Table 3: Comparison of different methods on the task of moving a distribution of smoke into the target region by applying forces outside the region.

| Method | Optimized quantity | Inside target (%) | Inference time (ms) |
|---|---|---|---|
| Straight trajectory | CFE | $89.5 \pm 2.6$ | $31.46 \pm 0.20$ |
| Staggered predictions | CFE, $\text{OP}_n$ | $99.22 \pm 0.15$ | $67.40 \pm 0.20$ |
| Iterative optim. | Control velocity | $82.1 \pm 7.3$ | $266.5 \cdot 10^3$ |

troduction of a hierarchical predictor-corrector architecture allowed the model to learn to reconstruct long sequences by treating the physical behavior on different time scales separately.

We have shown that using a differentiable solver greatly benefits the quality of solutions since the networks can learn how their decisions will affect the future. In our experiments, hierarchical inference schemes outperform traditional sequential agents because they can easily learn to plan ahead.

To model realistic environments, we have introduced observations to our pipeline which restrict the information available to the learning agent. While the PDE solver still requires full state information to run the simulation, this restriction does not apply when the agent is deployed.

While we do not believe that learning approaches will replace iterative optimization, our method shows that it is possible to learn representations of solution manifolds for optimal control trajectories using data-driven approaches. Fast inference is vital in time-critical applications and can also be used in conjunction with classical solvers to speed up convergence and ultimately produce better solutions.

## 8 ACKNOWLEDGEMENTS

This work was supported in part by the ERC Starting Grant *realFlow* (ERC-2015-StG-637014).

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

## A    IMPLEMENTATION DETAILS OF THE DIFFERENTIABLE PDE SOLVER

Our solver is publicly available at https://github.com/tum-pbs/PhiFlow, licensed as MIT. It is implemented via existing machine learning frameworks to benefit from the built-in automatic differentiation and to enable tight integration with neural networks.

For the experiments shown here we used the popular machine learning framework Tensor-Flow (Abadi et al., 2016). However, our solver is written in a framework-independent way and also supports PyTorch (Paszke et al., 2019). Both frameworks allow for a low-level NumPy-like implementation which is well suited for basic PDE building blocks. The following paragraphs outline how we implemented these building blocks and how they can be put together to solve the PDEs shown in Section 6.

**Staggered grids.** Many of the experiments presented in Section 6 use PDEs which track velocities. We adopt the marker-and-cell method (Harlow & Welch, 1965), storing densities in a regular grid and velocity in a staggered grid. Unlike regular grids, where all components are sampled at the centers of grid cells, staggered grids sample vector fields in a staggered form. Each vector component is sampled in the center of the cell face perpendicular to that direction. This sampling allows for an exact formulation of the divergence of a staggered vector field, decreasing discretization errors in many cases. On the other hand, it complicates operations that combine vector fields with regular fields such as transport or density-dependent forces.

We use staggered grids for the velocities in all of our experiments. The buoyancy operation, which applies an upward force proportional to the smoke density, interpolates the density to the staggered grid. For the transport, also called *advection*, of regular or staggered fields, we interpolate the staggered field to grid cell centers or face centers, respectively. These interpolations are implemented in TensorFlow using basic tensor operations, similar to the differential operators. We implemented all differential operators that act on vector fields to support staggered grids as well.

**Differential operators.** For the experiments outlined in this paper, we have implemented the following differential operators:

- Gradient of scalar fields in any number of dimensions, $\nabla x$
- Divergence of regular and staggered vector fields in any number of dimensions, $\nabla \cdot \boldsymbol{x}$
- Curl of staggered vector fields in 2D and 3D, $\nabla \times \boldsymbol{x}$
- Laplace of scalar fields in any number of dimensions, $\nabla^2 x$

All differential operators are local operations, i.e. they only act on a small neighbourhood of grid points. In the context of machine learning, this suggests implementing them as convolution operations with a fixed kernel. Indeed, all differential operators can be expressed this way and we have implemented some low-dimensional versions of them using this method.

This method does, however, scale poorly with the dimensionality of the physical system as the convolutional kernels pick up a large number of zeros, thus wasting computations. Therefore, we express n-dimensional differential operators using basic mathematical tensor operations.

Consider the gradient computation in 1D, which results in a staggered grid. Each resulting value is

$$(\nabla x)_i = x_i - x_{i-1},$$

assuming the result is staggered at the lower faces of each grid cell. This operation can be implemented as a 1D convolution with kernel $(-1, 1)$ or as a vector operation which subtracts the array from itself, shifted by one element.

In a low-dimensional setting, the convolution operation will be faster as it is highly optimized and can be executed on GPUs with one call. In higher dimensions, however, the vector-based version is faster and more practical because it avoids unnecessary computations and can be coded in a dimension-independent fashion. Both convolutions and basic mathematical operations are supported by all common machine learning frameworks, eliminating the need to implement custom gradient functions.

**Advection.** PDEs containing material derivatives can be solved using an advection step

$$f \leftarrow \text{Advect}[f, \boldsymbol{v}]$$

which moves each value of a field $f$ in the direction specified by a vector field $\boldsymbol{v}$. We implement the advection with semi-Lagrangian step (Stam, 1999) that looks back in time and supports regular and staggered vector fields.

To determine the advected value of a grid cell or face $x_{\text{target}}$, first $\boldsymbol{v}$ is interpolated to that point. Then the origin location is determined by following the vector backwards in time,

$$x_{\text{origin}} = x_{\text{target}} - \Delta t \cdot \boldsymbol{v}(x_{\text{target}}).$$

The final value is determined by linearly interpolating between the neighbouring grid cells around $x_{\text{origin}}$. All of these operations are, again, implemented using basic mathematical operations. Hence, gradients can be provided by the framework.

**Poisson problems.** Incompressible fluids, governed by the Navier-Stokes equations, are subject to the hard constraints $\nabla \cdot \boldsymbol{v} = 0$ and $\nabla \times p = 0$ where $p$ denotes the pressure. A numerical solver can achieve this by finding a $p$ such that these constraints are satisfied. This step is often called *Chorin Projection*, or *Helmholtz decomposition*, and is closely related to the fundamental theorem of vector calculus (von Helmholtz, 1858; Chorin, 1967). On a grid, solving for $p$ is equal to solving a Poisson problem, i.e. a system of $N$ linear equations, $\boldsymbol{A}p = \nabla \cdot u$ where $N$ is the total number of grid cells. The $(N \cdot N)$ matrix $A$ is sparse and its entries are located at predictable indices.

We numerically solve this Poisson problem with a conjugate gradient (CG) algorithm (Golub & Van Loan, 2012) that iteratively approximates $p$. Since hundreds of CG steps typically need to be performed for each Poisson solve, it is unfeasible to unroll this chain of iterations, and store all intermediate results in memory.

To ensure that the automatic differentiation chain is not broken, we instead solve the adjoint problem. For the pressure solve operation, the matrix $A$ is symmetric and positive-definite. This causes the adjoint problem to have the same mathematical form (McNamara et al., 2004) as the original problem. Therefore we implement the gradient for the pressure solve by performing a pressure solve on the gradient. We believe that this is a good example of leveraging the methodology of adjoint method optimizations (Giles & Pierce, 2000; Treuille et al., 2006; Pontryagin, 1962) within deep learning. With this formalism we arrive at a differentiable solver framework that closely integrates numerical methods for forward problems with support for inverse problems such as deep learning via the adjoint method.

**Solving Burger's equation and the Navier-Stokes equations.** Using the basic building blocks outlined above, solving the PDEs becomes relatively simple. Burger's equation involves an advection and a diffusion term which we evaluate independently.

$$\text{Solver}[u] = \text{Diffuse}[\text{Advect}[u, u]]$$

where we explicitly compute $\text{Diffuse}[u] = u + \nu \nabla^2 u$ with viscosity $\nu$. The advection is semi-Lagrangian with back-tracing as described above.

Solving the Navier-Stokes equations, typically comprises of the following steps:

- Transporting the density, $\rho \leftarrow \text{Advect}[\rho, \boldsymbol{v}]$
- Transporting the velocity, $\boldsymbol{v} \leftarrow \text{Advect}[\boldsymbol{v}, \boldsymbol{v}]$
- Applying diffusion if the viscosity is $\nu > 0$.
- Applying buoyancy force, $\boldsymbol{v} \leftarrow \boldsymbol{v} - \boldsymbol{\beta} \cdot \rho$ with buoyancy direction $\boldsymbol{\beta}$
- Enforcing incompressibility by solving for the pressure, $p \leftarrow \text{Solve}[Ap = \nabla \cdot \boldsymbol{v}]$, then $\boldsymbol{v} \leftarrow \boldsymbol{v} - \nabla p$

These steps are executed in this order to advance the simulation forward in time.

## B   COMPLEXITY OF EXECUTION SCHEMES

The staggered execution scheme recursively splits a sequence of length $n$ into smaller sequences, as depicted in Fig. 3b and Fig. 7a for $n = 8$. With each level of recursion depth, the sequence length

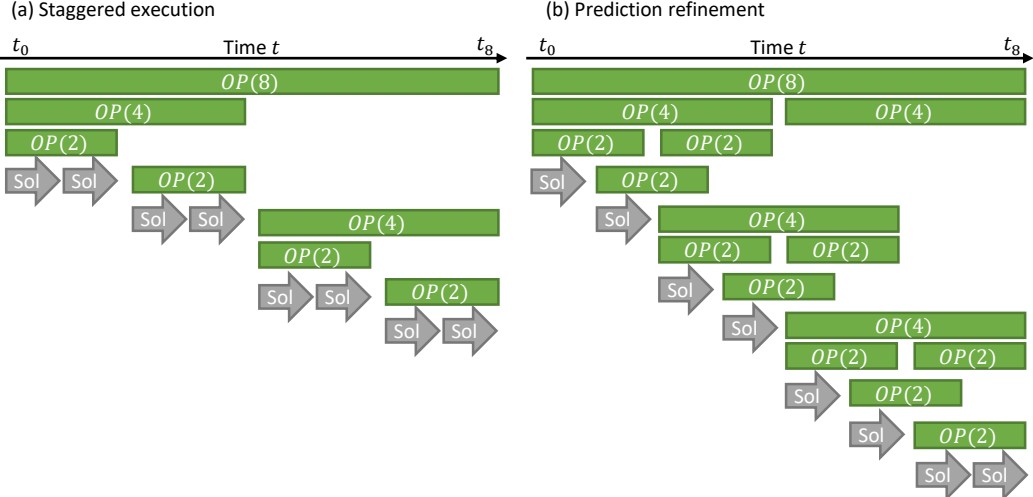

Figure 7: OP and CFE+Solver (Sol) executions for a sequence of length 8 performed by (a) the staggered execution scheme, (b) the prediction refinement scheme. The execution order is from top to bottom.

is cut in half and twice as many predictions need to be performed. The maximum depth depends on the sequence length $t_n - t_0$ and the time steps $\Delta t$ performed by the solver,

$$d_{\max} = \log_2 \left( \frac{t_n - t_0}{\Delta t} \right) - 1.$$

Therefore, the total number of predictions, equal to the number of OP evaluations, is

$$N_{\mathrm{OP}} = 1 + 2 + 4 + ... + n/2 = \sum_{k=0}^{d_{\max}} 2^k = n - 1.$$

The prediction refinement scheme performs more predictions, as can be seen in Fig. 7b. To understand the number of OP evaluations, we need to consider the recursive algorithm Reconstruct$[\boldsymbol{u}_0, \boldsymbol{o}_n, \boldsymbol{o}_{2n}]$, listed in Alg 1, that reconstructs a sequence or partial sequence of $n$ frames. For the first invocation, the last parameter $\boldsymbol{o}_{2n}$ is absent, but for subsequences, that is not necessarily the case. Each invocation performs one OP evaluation if $\boldsymbol{o}_{2n}$ is absent, otherwise three. By counting the sequences for which this condition is fulfilled, we can compute the total number of network evaluations to be

$$N_{\mathrm{OP}} = 3 \sum_{k=0}^{d_{\max}} 2^k - 2 \log_2(n) = 3n - 2 \log_2(n) - 3.$$

## C  NETWORK ARCHITECTURES AND TRAINING

All neural networks used in this work are based on a modified U-net architecture (Ronneberger et al., 2015). The U-net represents a typical multi-level convolutional network architecture with skip connections, which we modify by using residual blocks (He et al., 2016) instead of regular convolutions for each level. We slightly modify this basic layout for some experiments.

The network used for predicting observations for the fluid example is detailed in Tab. 4. The input to the network are two feature maps containing the current state and the target state. Zero-padding is applied to the input, so that all strided convolutions do not require padding. Next, five residual blocks are executed in order, each decreasing the resolution (1/2, 1/4, 1/8, 1/16, 1/32) while increasing the number of feature maps (4, 8, 16, 16, 16). Each block performs a convolution with kernel size 2 and stride 2, followed by two residual blocks with kernel size 3 and symmetric padding. Inside each

Table 4: Layers comprising the observation predictor network used in the direct fluid control experiment.

| Layer | Resolution | Feature Maps |
|---|---|---|
| Input | 128 | 2 |
| Pad | 159 | 2 |
| Strided convolution + 2x Residual block | 79 | 4 |
| Strided convolution + 2x Residual block | 39 | 8 |
| Strided convolution + 2x Residual block | 19 | 16 |
| Strided convolution + 2x Residual block | 9 | 16 |
| Strided convolution + 2x Residual block | 4 | 16 |
| 3x Residual block | 4 | 16 |
| Upsample + Concatenate | 8 | 32 |
| Convolution + 2x Residual block | 8 | 16 |
| Upsample + Concatenate | 16 | 32 |
| Convolution + 2x Residual block | 16 | 16 |
| Upsample + Concatenate | 32 | 24 |
| Convolution + 2x Residual block | 32 | 16 |
| Upsample + Concatenate | 64 | 20 |
| Convolution + 2x Residual block | 64 | 16 |
| Upsample + Concatenate | 128 | 18 |
| Convolution | 128 | 1 |

block, the number of feature maps stays constant. Three more residual blocks are executed on the lowest resolution of the bowtie structure, after which the decoder part of the network commences, translating features into spatial content.

The decoder works as follows: Starting with the lowest resolution, the feature maps are upsampled with linear interpolation. The upsampled maps and the output of the previous block of same resolution are then concatenated. Next, a convolution with 16 filters, a kernel size of 2 and symmetric padding, followed by two more residual blocks, is executed. When the original resolution is reached, only one feature map is produced instead of 16, forming the output of the network.

Depending on the dimensionality of the problem, either 1D or 2D convolutions are used. The network used for the indirect control task is modified in the following ways: (i) It produces two output feature maps, representing the velocity $(v_x, v_y)$. (ii) Four feature maps of the lowest resolution (4x4) are fed into a dense layer producing four output feature maps. These and the other feature maps are concatenated before moving to the upsampling stage. This modification ensures that the receptive field of the network is the whole domain.

All networks were implemented in TensorFlow (Abadi et al., 2016) and trained using the ADAM optimizer on an Nvidia GTX 1080 Ti. We use batch sizes ranging from 4 to 16. Supervised training of all networks converges within a few minutes, for which we iteratively decrease the learning rate from $10^{-3}$ to $10^{-5}$. We stop supervised training after a few epochs, comprising between 2000 and 10.000 iterations, as the networks usually converge within a fraction of the first epoch.

For training with the differentiable solver, we start with a decreased learning rate of $10^{-4}$ since the backpropagation through long chains is more challenging than training with a supervised loss. Optimization steps are also considerably more expensive since the whole chain needs to be executed, which includes a forward and backward simulation pass. For the fluid examples, an optimization step takes 1-2 seconds to complete for the 2D fluid problems. We let the networks run about 100.000 iterations, which takes between one and two days for the shown examples.

## D    DETAILED DESCRIPTION AND ANALYSIS OF THE EXPERIMENTS

In the following paragraphs, we give further details on the experiments of Section 6.

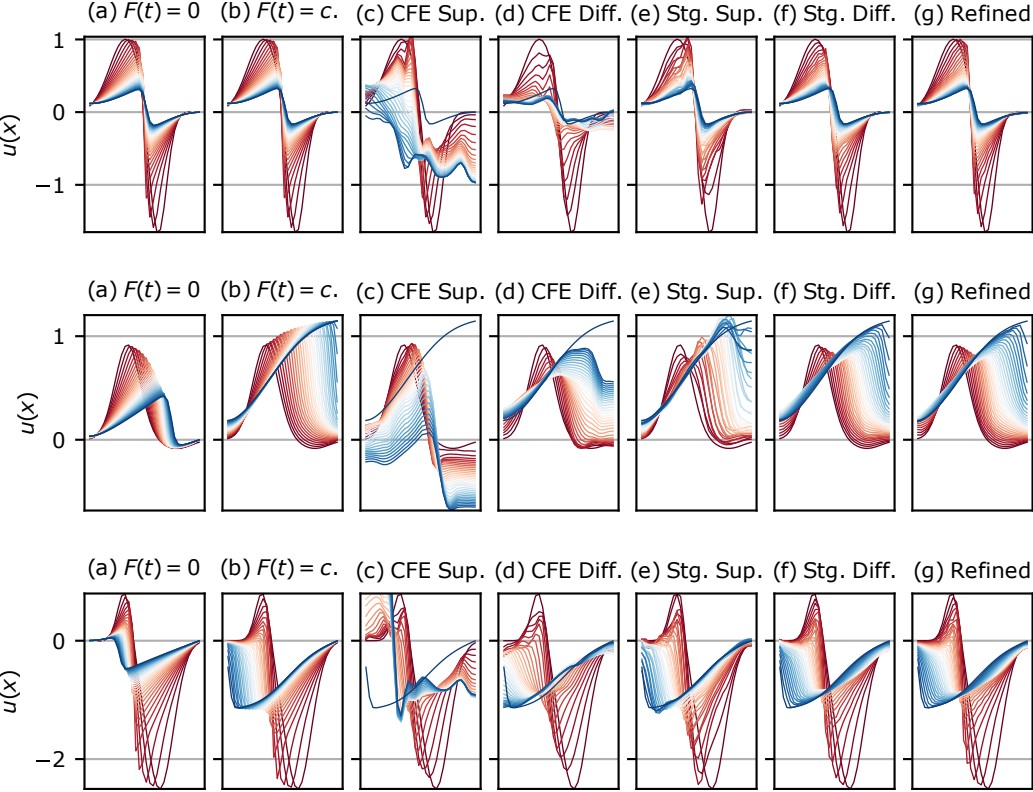

Figure 8: Trajectories for example control tasks using Burger's equation. The initial state is plotted in red, the target state in blue. (a) Natural evolution, (b) Ground truth trajectory generated with constant force, (c) CFE with supervised loss, (d) CFE with differentiable physics loss, (e) Reconstruction with supervised loss, (f) Reconstruction with differentiable physics loss and staggered execution, (g) Reconstruction with differentiable physics loss and prediction refinement.

## D.1 BURGER'S EQUATION

For this experiment, we simulate Burger's equation (Eq. 6) on a one-dimensional grid with 32 samples over a course of 32 time steps. The typical behavior of Burger's equation in 1D exhibits shock waves that move in $+x$ or $-x$ direction for $u(x) > 0$ or $u(x) < 0$, respectively. When opposing waves clash, they both weaken until only the stronger wave survives and keeps moving. Examples are shown in Figs. 4a and 8a.

All 32 samples are observable and controllable, i.e. $o(t) = u(t)$. Thus, we can enforce that all trajectories reach the target state exactly by choosing the force for the last step to be

$$F(t_{n-1}) = \frac{o^* - u(t_{n-1})}{\Delta t}.$$

To measure the quality of a solution, it is therefore sufficient to consider the applied force $\int_{t_0}^{t_*} |F(t)| \, dt$ which is detailed for the tested methods in Table 1.

**Network training.** Both for the CFE chains as well as for the observation prediction models, we use the same network architecture, described in Appendix C. We train the networks on 3600 randomly generated scenes with constant driving forces, $F(t) = \text{const}$. The examples are initialized with two Gaussian waves of random amplitude, size and position, set to clash in the center. In each time step, a constant Gaussian force with the same randomized parameters is applied to the system to steer it away from its natural evolution. Constant forces have a larger impact on the evolution than temporally varying forces since the effects of temporally varying forces can partly cancel out over time. The ground truth sequence can therefore be regarded as a near-perfect but not necessarily

optimal trajectory. Figs. 4d and 8b display such examples. The same trajectories, without any forces applied, are shown in sub-figures (a) for comparison.

We pretrain all networks (OPs or CFE, depending on the method) with a supervised observation loss,

$$L_o^{\text{sup}} = \left| \text{OP}[o(t_i), o(t_j)] - u^{\text{GT}}\left(\frac{t_i + t_j}{2}\right) \right|^2. \tag{8}$$

The resulting trajectory after supervised training for the CFE chain is shown in Figure 4b and Figure 8c. For the observation prediction models, the trajectories are shown in Figure 4c and Figure 8e.

After pretraining, we train all OP networks end-to-end with our objective loss function (see Eq. 4), making use of the differentiable solver. For this experiment, we choose the mean squared difference for the observation loss function:

$$L_o^* = |o(u(t_*)) - o^*|^2. \tag{9}$$

We test both the staggered execution scheme and the prediction refinement scheme, shown in Figure 8f and Figure 8g.

**Results.** Table 1 compares the resulting forces inferred by different methods. The results are averaged over a set of 100 examples from the test set which is sampled from the same distribution as the training set. The CFE chains both fail to converge to $o^*$. While the differentiable physics version manages to produce a $u_{n-1}$ that resembles $o^*$, the supervised version completely deviates from an optimal trajectory. This shows that learning to infer the control force $F(t_i)$ only from $u(t_i)$, $o^*$ and $t$ is very difficult as the model needs to learn to anticipate the physical behavior over any length of time.

Compared to the CFE chains, the hierarchical models require much less force and learn to converge towards $o^*$. Still, the supervised training applies much more force to the system than required, the reasons for which become obvious when inspecting Figure 4b and Fig. 8e. While each state seems close to the ground truth individually, the control oscillates undesirably, requiring counter-actions later in time.

The methods using the differentiable solver significantly outperform their supervised counterparts and exhibit an excellent performance that is very close the ground truth solutions in terms of required forces. On many examples, they even reach the target state with less force than was applied by the ground truth simulation. This would not be possible with the supervised loss alone, but by having access to the gradient-based feedback from the differentiable solver, they can learn to find more efficient trajectories with respect to the objective loss. This allows the networks to learn applying forces in different locations that make the system approach the target state with less force.

Figure 4e and Fig.8f,g show examples of this. The ground truth applies the same force in each step, thereby continuously increasing the first sample $u(x = 0)$, and the supervised method tries to imitate this behavior. The governing equation then slowly propagates $u(x = 0)$ in positive $x$ direction since $u(x = 0) > 0$. The learning methods that use a differentiable solver make use of this fact by applying much more force $F(x = 0) > 0$ at this point than the ground truth, even overshooting the target state. Later, when this value had time to propagate to the right, the model corrects this overshoot by applying a negative force $F(x = 0) < 0$. Using this trick, these models reach the target state with up to 13% less force than the ground truth on the sequence shown in Figure 4.

Figure 9 analyzes the variance of inferred forces. The supervised methods often fail to properly converge to the target state, resulting in large forces in the last step, visible as a second peak in the supervised CFE chain. The formulation of the loss (Eq. 3) suppresses force spikes. In the solutions inferred by our method, the likelihood of large forces falls off multi-exponentially as a consequence. This means that large forces are exponentially rare, which is the expected behavior given the L2 regularizer from Eq. 3.

We also compare our results to a single-shooting baseline which is able to find near-optimal solutions at the cost of higher computation times. The classic optimization uses the ADAM optimizer with a learning rate of 0.01 and converges after around 300 iterations. To reach the quality of the staggered prediction scheme, it requires only around 60 iterations. This quick convergence can be explained by the relatively simple setup that is dominated by linear effects. Therefore, the gradients are stable, even when propagated through many frames.

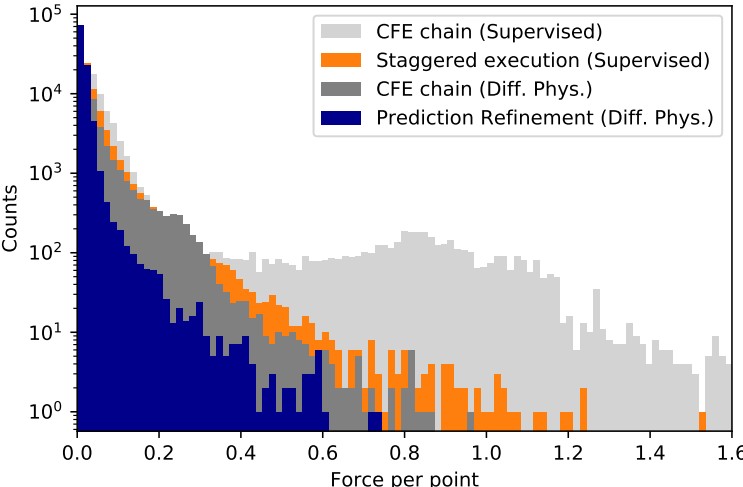

Figure 9: Histogram comparing the frequency of force strengths of different methods, summed over 100 examples from the Burger's experiment.

The computation times, shown in Tab. 1, were recorded on a single GTX 1080 Ti. We run 100 examples in parallel to reduce the relative overhead caused by GPU instruction queuing. For the network-based methods, we average the inference time over 100 runs. We perform 10 runs for the optimization methods.

### D.2    INCOMPRESSIBLE FLUID FLOW

The incompressible Navier-Stokes equations model dynamics of fluids such as water or air, which can develop highly complex and chaotic behavior. The phenomenon of turbulence is generally seen as one of the few remaining fundamental and unsolved problems of classical physics. The challenging nature of the equations indicates that typically a very significant computational effort and a large number of degrees of freedom are required to numerically compute solutions. Here, we target an incompressible two-dimensional gas with viscosity $\nu$, described by the Navier-Stokes equations for the velocity field $\boldsymbol{v}$. We assume a constant fluid density throughout the simulation, setting $\rho_f = \mathrm{const.} \equiv 1$. The gas velocity is controllable and, according to Eq. 1, we set

$$\mathcal{P}(\boldsymbol{v}, \nabla\boldsymbol{v}) = -(\boldsymbol{v} \cdot \nabla)\boldsymbol{v} + \nu\nabla^2\boldsymbol{v} - \frac{\nabla p}{\rho_f}$$

subject to the hard constraints $\nabla \cdot \boldsymbol{v} = 0$ and $\nabla \times p = 0$. For our experiments, we target fluids with low viscosities, such as air, and set $\nu = 0$ in the equation above as the transport steps implicitly apply numerical diffusion that is on average higher than the targeted one. For fluids with a larger viscosity, the Poisson solver outlined above for computing $p$ could be used to implicitly solve a vector-valued diffusion equation for $\boldsymbol{v}$.

However, incorporating a significant amount of viscosity would make the control problem easier to solve for most cases, as viscosity suppresses small scale structures in the motion. Hence, in order to create a challenging environment for training our networks, we have but a minimal amount of diffusion in the physical model.

In addition to the velocity field $\boldsymbol{v}$, we consider a smoke density distribution $\rho$ which moves passively with the fluid. The evolution of $\rho$ is described by the equation $\partial\rho/\partial t = -v \cdot \nabla\rho$. We treat the velocity field as hidden from observation, letting only the smoke density be observed, i.e. $o(t) = \rho(t)$. We stack the two fields as $u = (v, \rho)$ to write the system as one PDE, compatible with Eq. 1.

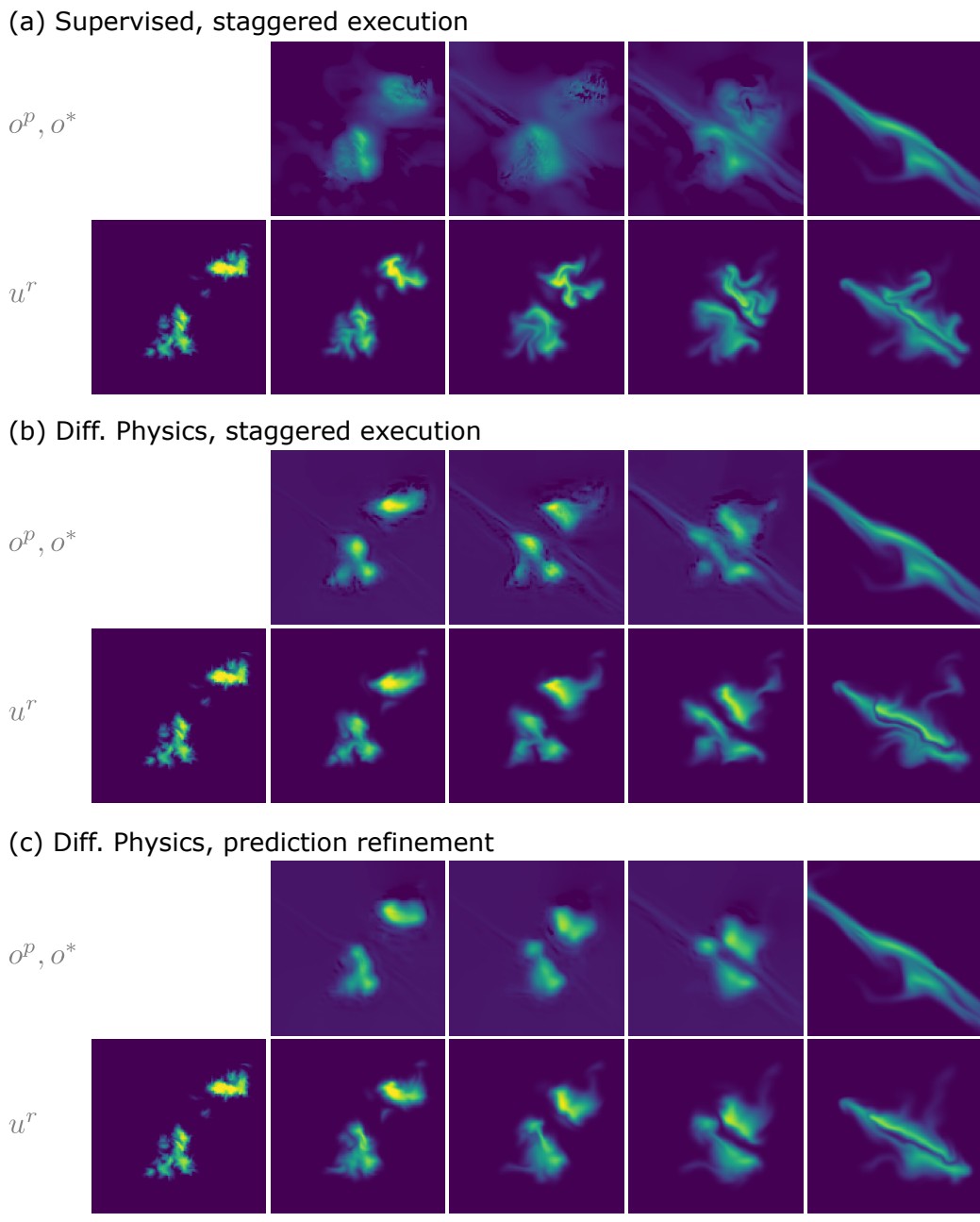

Figure 10: Reconstruction of an example natural flow sequence from the test set. The predictions $o^p$ are plotted above the reconstructions $u$. The target state $o^*$ is shown in the last column of the predictions.

For the OP and CFE networks, we use the 2D network architecture described in Appendix C. Instead of directly generating the velocity update in the CFE network for this problem setup, we make use of stream functions (Lamb, 1932). Hence, the CFE network outputs a vector potential $\Phi$ of which the curl $\nabla \times \Phi$ is used as a velocity update. This setup numerically simplifies the incompressibility condition of the Navier-Stokes equations but retains the same number of effective control parameters.

**Datasets.** We generate training and test datasets for two distinct tasks: flow reconstruction and shape transition. Both datasets have a resolution of $128 \times 128$ with the velocity fields being sampled in staggered form (see Appendix A). This results in over 16.000 effective continuous control parameters that make up the control force $\boldsymbol{F}(t_i)$ for each step $i$.

The flow reconstruction dataset is comprised of ground-truth sequences where the initial states $(\rho_0, \boldsymbol{v}_0)$ are randomly sampled and then simulated for 64 time steps. The resulting smoke density is then taken to be the target state, $\boldsymbol{o}^* \equiv \rho^* = \rho^{\mathrm{sim}}(t_{64})$. Since we use fully convolutional networks for both CFE and OPs, the open domain boundary must be handled carefully. If smoke was lost from the simulation, because it crossed the outer boundary, a neural network would see the smoke simply vanish unless it was explicitly given the domain size as input. To avoid these problems, we run the simulation backwards in time and remove all smoke from $\rho_0$ that left the simulation domain.

For the shape transition dataset, we sample initial and target states $\rho_0$ and $\rho_*$ by randomly choosing a shape from a library containing ten basic geometric shapes and placing it at a random location inside the domain. These can then be used for reconstructing sequences of any length $n$. For the results on shape transition presented in section 6, we choose $n = 16$ because all interesting behavior can be seen within that time frame. Due to the linear interpolation used in the advection step (see Appendix A), both $\rho$ and $\boldsymbol{v}$ smear out over time. This numerical limitation makes it impossible to match target states exactly in this task as the density will become blurry over time. While we could generate ground-truth sequences using a classical optimizer, we refrain from doing so because (i) these trajectories are not guaranteed to be optimal and (ii) we want to see how well the model can learn from scratch, without initialization.

**Training.** We pretrain the CFE on the natural flow dataset with a supervised loss,

$$L_{\mathrm{sup}}^{\mathrm{CFE}}(\boldsymbol{u}(t)) = |\boldsymbol{v}_{\boldsymbol{u}(t)} + \boldsymbol{F}(t) - \boldsymbol{v}^*(t)|^2$$

where $\boldsymbol{v}^*(t)$ denotes the velocity from ground truth sequences. This supervised training alone constitutes a good loss for the CFE as it only needs to consider single-step intervals $\Delta t$ while the OPs handle longer sequences. Nevertheless, we found that using the differentiable solver with an observation loss,

$$L_{\boldsymbol{o}}^{\mathrm{CFE}} = |B_r(\boldsymbol{o}^*) - B_r\left(\mathrm{Solver}[\boldsymbol{u} + \mathrm{CFE}[\boldsymbol{u}, \boldsymbol{o}^*]]\right)|^2,$$

further improves the accuracy of the inferred force without sacrificing the ground truth match. Here $B_r(x)$ denotes a blur function with a kernel of the form $\frac{1}{1+x/r}$. The blur helps make the gradients smoother and creates non-zero gradients in places where prediction and target do not overlap. During training, we start with a large radius of $r = 16 \, \Delta x$ for $B_r$ and successively decrease it to $r = 2 \, \Delta x$. We choose $\alpha$ such that $L_{\boldsymbol{F}}$ and $L_{\boldsymbol{o}}^*$ are of the same magnitude when the force loss spikes (see Fig. 15).

After the CFE is trained, we successively train the OPs starting with the smallest time scale. For the OPs, we train different models for natural flow reconstruction and shape transition, both based on the same CFE model. We pre-train all OPs independently with a supervised observation loss before jointly training them end-to-end with objective loss function (Eq. 4) and the differentiable solver to find the optimal trajectory. We use the OPs trained with the staggered execution scheme as initialization for the prediction refinement scheme. The complexity of solving the Navier-Stokes equations over many time steps in this example requires such a fully supervised initialization step. Without it, this setting is so non-linear that the learning process does not converge to a good solution. Hence, it illustrates the importance of combining supervised and unsupervised (requiring differentiable physics) training for challenging learning objectives.

A comparison of the different losses is shown in Fig. 10. The predictions, shown in the top rows of each subfigure, illustrate the differences between the three methods. The supervised predictions, especially the long-term predictions (central images), are blurry because the network learns to average over all ground truth sequences that match the given initial and target state. The differentiable

(a) Predicted trajectory

(b) Reconstructed trajectory

Figure 11: Reconstruction of multiple shapes using prediction refinement. The CFE is trained on the natural flow dataset and OPs are trained on single-shape transitions. The predictions of all shapes are added and passed to the CFE as one prediction.

physics solver largely resolves this issue. The predictions are much sharper but the long-term predictions still do not account for short-term deviations. This can be seen in the central prediction of Fig. 10b which shows hints of the target state $o^*$, despite the fact that the actual reconstruction $u$ cannot reach that state at that time. The refined prediction, shown in subfigure (c), is closer to $u$ since it is conditioned on the previous reconstructed state.

In the training data, we let the network transform one shape into another at a random location. The differentiable solver and the long-term intuition provided by our execution scheme make it possible to train networks that can infer accurate sequences of control forces. In most cases, the target shapes are closely matched. As our networks infer sequences over time, we refer readers to the supplemental material (https://ge.in.tum.de/publications/2020-iclr-holl), which contains animations of additional sequences.

**Generalization to multiple shapes.** Splitting the reconstruction task into prediction and correction has the additional benefit of having full access to the intermediate predictions $o^p$. These model real states of the system so classical processing or filter operations can be applied to them as well. We demonstrate this by generalizing our method to $m > 1$ shapes that evolve within the same domain. Figure 11 shows an example of two weakly-interacting shape transitions. We implement this by executing the OPs independently for each transition $k \in \{1, 2, ...m\}$ while inferring the control force $\mathbf{F}(t)$ on the joint system. This is achieved by adding the predictions of the smoke density $\rho$ before passing it to the CFE network, $\tilde{o}^p = \sum_{k=1}^{m} o_k^p$. The resulting force is then applied to all sequences individually so that smoke from one transition does not end up in another target state. Using this scheme, we can define start and end positions for arbitrarily many shapes and let them evolve together.

**Evaluation of force strengths** The average force strengths are detailed in Tab. 2 while Figure 12 gives a more detailed analysis of the force strengths. As expected from using a L2 regularizer on the force, large values are exponentially rare in the solutions inferred from our test set. None of the hierarchical execution schemes exhibit large outliers. The prediction refinement requires the least amount of force to match the target, slightly ahead of the staggered execution trained with the same loss. The supervised training produces trajectories with reduced continuity that result in larger forces being applied.

### D.3 INCOMPRESSIBLE FLUID WITH INDIRECT CONTROL

As a fourth test environment, we target a case with increased complexity, where the network does not have the means anymore to directly control the full fluid volume. Instead, the network can only apply forces in the peripheral regions, with a total of more than 5000 control parameters per step. The obstacles prevent fluid from passing through them and the domain is enclosed with solid boundaries from the left, right and bottom. This leads to additional hard constraints and interplays between constraints in the physical model, and as such provides an interesting and challenging test case for our method. The domain has three target regions (*buckets*) separated by walls at the top of the domain, into which a volume of smoke should be transported from any position in the center

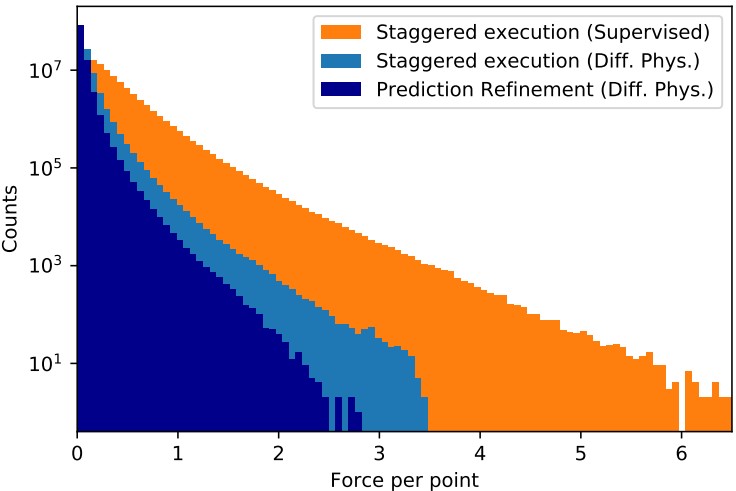

Figure 12: Histogram comparing the frequency of force strengths applied in the direct fluid control experiment on the natural flow dataset, summed over 100 examples.

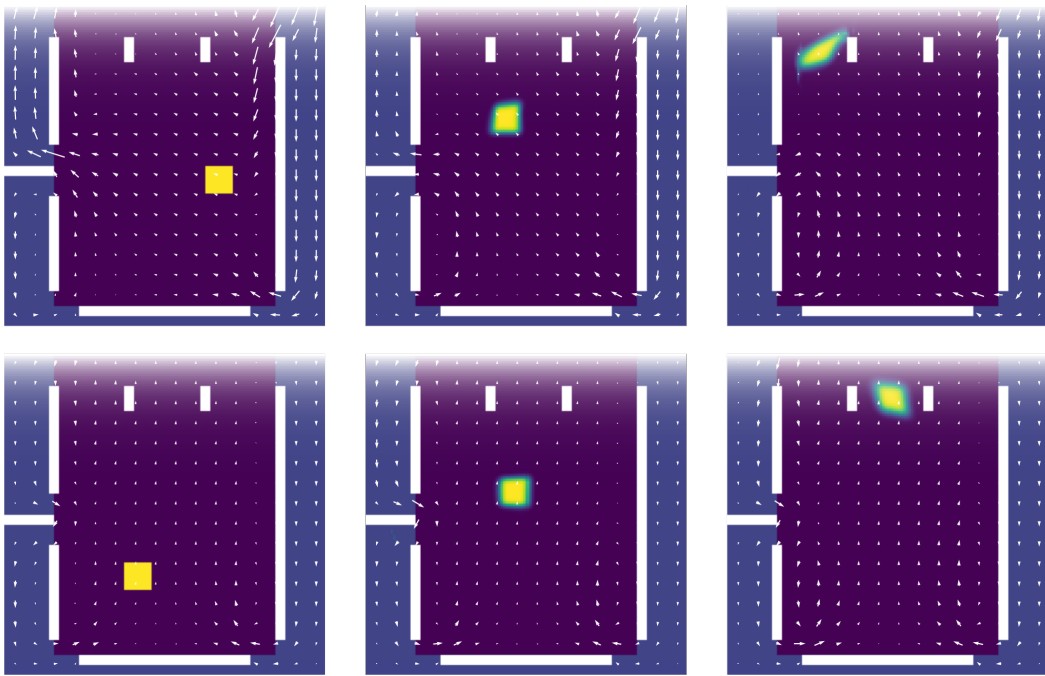

Figure 13: Two reconstructed trajectories from the test set of the indirect smoke control problem.

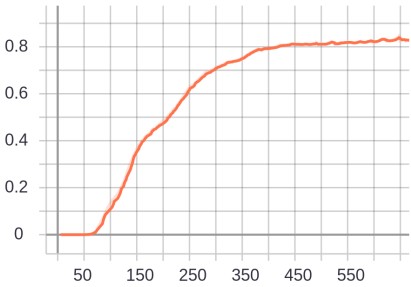

Figure 14: Iterative optimization of the indirect incompressible fluid control problem. The graph shows the fraction of smoke that ends up in the correct bucket vs number of optimization steps, averaged over 10 examples.

part. Both initial position and the target bucket are randomized for our training set of 3600 examples and test set of 100 examples. Each sequence consists of 16 time steps.

In this case the control is indirect since the smoke density lies outside the controlled area at all times. Only the incompressibility condition allows the network to influence the velocity outside the controlled area. This forces the model to consider the global context and synchronize a large number of parameters to create a desired flow field. The requirement of complex synchronized force fields makes generating reliable training data difficult, as manual or random sampling is unlikely to produce a directed velocity field in the center. We therefore skip the pretraining process and directly train the CFE using the differentiable solver, while the OP networks are trained as before with $r = 2\,\Delta x$.

To evaluate how well the learning method performs, we measure how much of the smoke density ends up inside the buckets and how much force was applied in total. For reference, we replace the observation predictions with an algorithm that moves the smoke towards the bucket in a straight line. Averaged over 100 examples from the test set, the resulting model manages to put $89\% \pm 2.6\%$ of the smoke into the target bucket. In contrast, the model trained with our full algorithm moves $99.22\% \pm 0.15\%$ of the smoke into the target buckets while requiring $19.1\% \pm 1.0\%$ less force.

We also compare our method to an iterative optimization which directly optimizes the control velocities. We use the ADAM optimizer with a learning rate of 0.1. Despite the highly non-linear setup, the gradients are stable enough to quickly let the smoke flow in the right direction. Fig. 14 shows how the trajectories improve during optimization. After around 60 optimization steps, the smoke distribution starts reaching the target bucket in some examples. Over the next 600 iterations, it converges to a a configuration in which $82.1 \pm 7.3$ of the smoke ends up in the correct bucket.

### D.4 COMPARISON TO SHOOTING METHODS

We compare the sequences inferred by our trained models to classical shooting optimizations using our differentiable physics solver to directly optimize $F(t)$ with the objective loss $L$ (Eq. 4) for a single input. We make use of stream functions (Lamb, 1932), as in the second experiment, to ensure the incompressibility condition is fulfilled. For this comparison, the velocities of all steps are initialized with a normal distribution with $\mu = 0$ and $\sigma = 0.01$ so that the initial trajectory does not significantly alter the initial state, $u(t) \approx u(t_0)$.

We first show how a simple single-shooting algorithm (Zhou et al., 1996) fares with our Navier-Stokes setup. When solving the resulting optimization problem using single-shooting, strong artifacts in the reconstructions can be observed, as shown in Figure 17a. This undesirable behavior stems from the nonlinearity of the Navier-Stokes equations, which causes the gradients $\Delta u \gg 0$ to become noisy and unreliable when they are recurrently backpropagated through many time steps. Unsurprisingly, the single-shooting optimizer converges to a undesirable local minimum.

As single-shooting is well known to have problems with non-trivial problem settings, we employ a multi-scale shooting (MS) method (Hartmann et al., 2014). This solver first computes the trajectory on a coarsely discretized version of the problem before iteratively refining the discretization. For

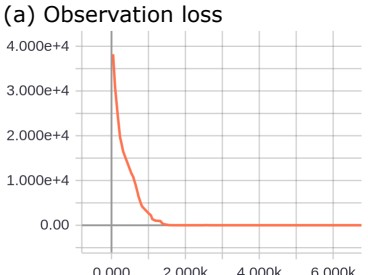 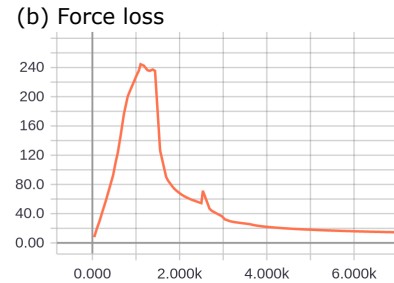

Figure 15: Mean convergence curves of the adjoint method optimization for 100 shape transitions.

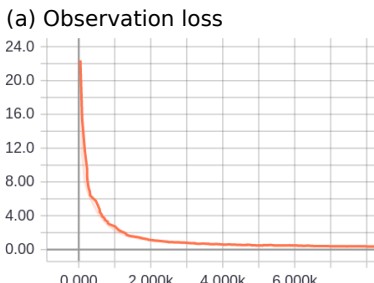 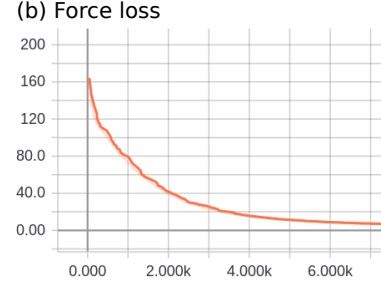

Figure 16: Mean convergence curves of the adjoint method optimization for 100 shape transitions, taking the reconstruction from the refinement scheme as initial guess.

the first resolution, we use 1/16 of the original width and height which both reduces the number of control parameters and reduces nonlinear effects from the physics model. By employing an exponential learning rate decay, this multi-scale optimization converges reliably for all examples. We use the ADAM optimizer to compute the control variable updates from the gradients of the differentiable Navier-Stokes solver.

An averaged set of representative convergence curves for this setup is shown in Figure 15. The objective loss (Eq. 4) is shown in its decomposed state as the sum of the observation loss $L_o^*$, shown in Figure 15a, and the force loss $L_F$, shown in Figure 15b. Due to the initialization of all velocities with small values, the force loss starts out small. For the first 1000 iteration steps, $L_o^*$ dominates which causes the system to move towards the target state $o^*$. This trajectory is not ideal, however, as more force than necessary is applied. Once observation loss and force loss are of the same magnitude, the optimization refines the trajectory to use less force.

We found that the trajectories predicted by our neural network based method correspond to performing about 1500 steps with the MS optimization while requiring less tuning. Reconstructions of the same example are compared in Figure 17. Performing the MS optimization up to this point took 131 seconds on a GTX 1080 Ti graphics card for a single 16-frame sequence while the network inference ran for 0.5 seconds. For longer sequences, this gap grows further because the network inference time scales with $\mathcal{O}(n)$. This could only be matched if the number of iterations for the MS optimization scaled with $O(1)$, which is not the case for most problems. These tests indicate that our model has successfully internalized the behavior of a large class of physical behavior, and can exert the right amount of force to reach the intended goal. The large number of iterations required for the single-case shooting optimization highlights the complexity of the individual solutions.

Interestingly, the network also benefits from the much more difficult task to learn a whole manifold of solutions: comparing solutions with similar observation loss for the MS algorithm and our network, the former often finds solutions that are unintuitive and contain noticeable detours, e.g., not taking a straight path for the density matching examples of Fig. 5. In such situations, our network benefits from having to represent the solution manifold, instead of aiming for single task optimizations. As the solutions are changing relatively smoothly, the complex task effectively regularizes the inference of new solutions and gives the network a more global view. Instead, the shooting optimiza-

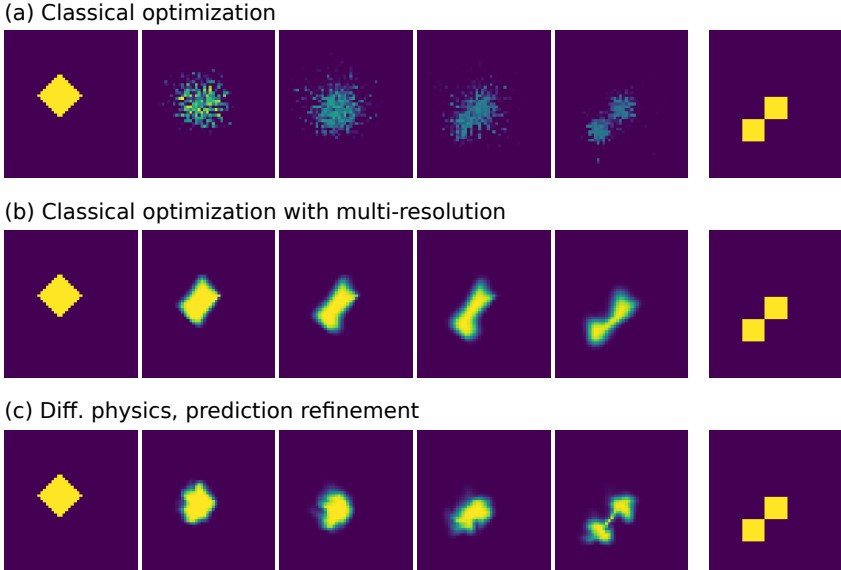

Figure 17: Example reconstruction of a shape transition. (a) Direct shooting optimization, 2300 iterations, (b) multi-scale shooting optimization, 1500 iterations, (c) output of our neural network based method with prediction refinement. Our model infers the shown solution in a single pass, and generalizes to a large class of inputs.

tions have to purely rely on local gradients for single-shooting or manually crafted multi-resolution schemes for MS.

Our method can also be employed to support the MS optimization by initializing it with the velocities inferred by the networks. In this case, shown in Figure 16, both $L_o^*$ and $L_F$ decrease right from the beginning, similar to the behavior in Figure 15 from iteration 1500 on. The reconstructed trajectory from the neural-network-based method is so close to the optimum that the multi-resolution approach described above is not necessary.

## D.5 ADDITIONAL RESULTS

In Fig. 18, we provide a visual overview of a sub-set of the sequences that can be found in the supplemental materials. It contains 16 randomly selected reconstructions for each of the natural flow, the shape transitions, and the indirect control examples. In addition, the supplemental material, available at https://ge.in.tum.de/publications/2020-iclr-holl, highlights the differences between unsupervised, staggered, and refined versions of our approach.

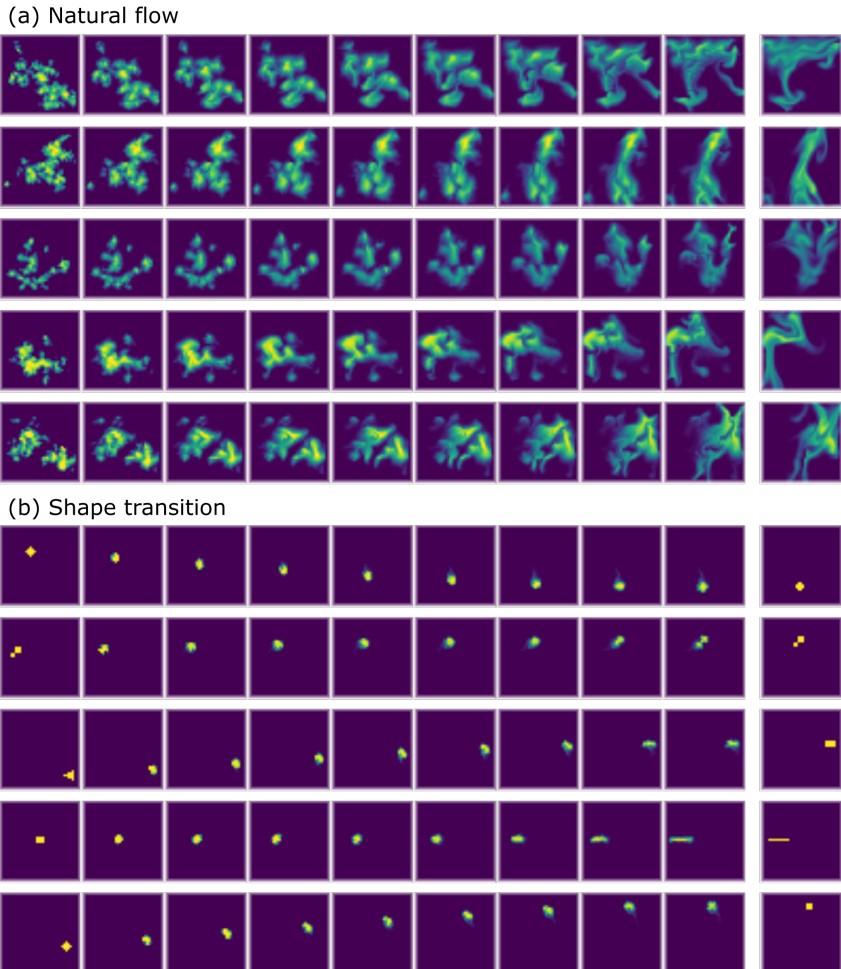

Figure 18: Five additional sequences from the test sets of the natural flow and shape transition setups. The first nine frames contain frames from our reconstruction. The far right image shows the target.

