# OpenReview forum: "Learning to Control PDEs with Differentiable Physics"
_ICLR.cc/2020/Conference — Accept (Spotlight)_

### Official Review · AnonReviewer1 · 2019-10-22
**Official Blind Review #1**

**Rating:** 6

**Review:**

In this paper, the authors outline a method for system control utilizing an "agent" formed by two neural networks and utilizing a differentiable grid-based PDE solver (assuming the PDE describing the system is known). The agent is split into a control force estimator (CFE) which applies a force to advance the state of the controlled system, and an observation predictor (OP) which predicts the trajectory needed to reach the target state. The objective is to reach the target state with minimal total amount of applied force. The order of CFE and OP calls is discussed and the importance of keeping the trajectory predictions conditioned on the actual previous state of the system, so that errors from previous steps can be taken into account.

Three application examples are discussed: Burger's equation (1D), and incompressible flow (2D) with direct and indirect control. In all cases the proposed scheme of "prediction refinement" leads to better or comparable results than standard iterative optimization, and is much more computationally efficient (at inference time, not taking into account the cost of training).

The paper presents an interesting mix of neural networks and traditional PDE solvers for system control, and I vote for acceptance. An additional advantage of the paper is the authors' promise to open source their differentiable PDE solver implemented in TensorFlow, which should make it easy for others to build upon their work. The text is easy to read, but quite verbose, with many of the technical details relegated to the (sizeable) appendices. I would recommend trying to trim it down where possible (for instance, the description of the U-nets could be more compact, perhaps in table form; 2nd paragraph of the background section seems a bit out of context and could probably be omitted, etc).

Questions and suggestions for improvements:

* What form of L_o^* and alpha was used in all the experiments?
* It looks like pretraining was used for all cases except for the most challenging one with indirect control. Was it truly necessary for the simpler experiments?
* Improve naming consistency. It looks like "differentiable physics" and "differentiable solver" are used for the same thing in different parts of the paper. My recommendation would be to use the latter term everywhere.
* How many time steps are used in the indirect control experiment?
* IIUC, the optimization of Eq. 3 is always done end-to-end. Have any experiments been done to estimate how many time steps can be reliably handled by the proposed procedure before the optimization problem becomes too hard?


**Experience Assessment:**

I have read many papers in this area.

**Review Assessment: Checking Correctness Of Derivations And Theory:**

I assessed the sensibility of the derivations and theory.

**Review Assessment: Checking Correctness Of Experiments:**

I assessed the sensibility of the experiments.

**Review Assessment: Thoroughness In Paper Reading:**

I read the paper at least twice and used my best judgement in assessing the paper.

---

> ### Author Response · Authors · 2019-11-12
> **Response to Review #1**
>
> Thank you very much for your positive review of our manuscript and the helpful and constructive feedback. We have addressed your comments in a revised version of our paper, which we uploaded along with this comment. We have highlighted major changes in blue.
>
>
> > “The text is [...] quite verbose. [...] I would recommend trying to trim it down where possible.”
>
> We removed the second paragraph of the Background section, and added a table detailing the network architecture layer by layer to make the description easier to follow.
>
>
> > “What form of L_o^* and alpha was used in all the experiments?”
>
> For the observation loss, we used the squared difference for our Burger’s experiment and the squared difference of spatially blurred density fields for the smoke experiments. We added the missing information to the appendix.
>
> Alpha was set to 10^-4 in our fluid experiments but this value is of limited significance as the loss values have different physical dimensions and require some arbitrary normalization. We chose alpha such that force and observation loss are of the same magnitude when the force loss spikes (see Fig. 13 / 15 in revised version).
>
>
> > “Was [pretraining] truly necessary for the simpler experiments?”
>
> For the Burger’s example, pretraining was not necessary due to the relatively low complexity.
>
> For the direct smoke control, the CFE could also be trained directly with the differentiable physics loss. However, the OPs (especially the longer-term ones) would not converge to a good solution without pretraining as the gradients become increasingly unstable over time in such a highly non-linear system.
>
> While one could increase training time or add more terms to the loss function to help in that regard, one advantage of pretraining is that it is very fast. All OPs can be trained in approximately the same amount of time. With the differentiable physics loss, the training time rises approximately linearly with the number of frames. Additionally, the pretraining does not need to execute any solver steps so it is much faster than training with the differentiable physics loss.
>
>
> > “ It looks like "differentiable physics" and "differentiable solver" are used for the same thing in different parts of the paper.”
>
> In our paper, differentiable physics refers to the optimization (e.g. differentiable physics loss) technique while the differentiable soler refers to the algorithm that computes the forward physics plus corresponding gradients. A differentiable solver is required to optimize for the differentiable physics loss. We now made this distinction clearer in the Preliminaries section.
>
>
> > “How many time steps are used in the indirect control experiment?”
>
> 16 time steps (17 frames). This could be increased with more training time, however. We added this number to the appendix.
>
>
> > “IIUC, the optimization of Eq. 3 is always done end-to-end. Have any experiments been done to estimate how many time steps can be reliably handled by the proposed procedure before the optimization problem becomes too hard?”
>
> Correct. After pretraining, we optimize for Eq. 3 end-to-end. The fact that gradients can become increasingly unstable over time is inherent to all gradient-based optimization algorithms, e.g., as discussed by Pascanu et al., "On the difficulty of training recurrent neural networks", ICML 2013.
>
> Our method is affected by this in much the same way. However, it does have two advantages over iterative optimization:
> 1. Data. Our models optimize for many examples at the same time. The gradients of these examples are averaged which could make them more reliable.
> 2. Pretraining helps find a good initial guess. The closer a solution is to the optimum, the more stable the gradients become.

---

### Official Review · AnonReviewer3 · 2019-10-25
**Official Blind Review #3**

**Rating:** 8

**Review:**

## Summary

The authors propose a method for training physical systems whose behavior is governed by partial differential equations. They consider situations where only partial observations available, and where control is indirect.

Indirectly controlling physical systems is a very important problem with applications throughout engineering. In fact, much of the field of robotics can be described in these terms. The authors employ a number of interesting methods, including a predictor-corrector framework and the adjoint sensitivity method for differentiating through differential equation solvers.

The paper is generally very clear, organized and well written. There are only a few places where I think clarification is needed (see detailed comments below). I also have a few questions about the losses and training procedure. On the whole, I think the paper is inventive, well-written and potentially very impactful. I think it would be a great addition to ICLR.


## Clarifications

* Page 4: I found the statement "an agent trained in supervised fashion will then learn to average over the modes instead of picking one of them" a little confusing. Could you clarify the reasoning here?
* Page 5: I think the description of predictor-corrector could be clearer. In particular, I found the phrase "the correction uses o(t + ∆t) to obtain o(t + ∆t)" unclear.
* Page 8: Could you add a description of what is observable to the body of the paper (I see it is included in the supplement)?
* Page 8: I think ∇p needs to be divided by density in your NS equation, right?
* Page 8 - 9: Is there a limit on the size of the force that can be applied at any point? I know the total force is penalized, but what about the maximum force applied at any point?


## Losses and Training

* I think "differentiable physics" losses need a more detailed explanation in the body of the paper.
* In the supplement, it is defined using B_r, but I don't think B_r is defined.
* It seems like the differential physics loss requires a differential solver (in this case, for Burger/Navier-Stokes). If I have understood this correctly, I think this needs to be discussed in the body of the paper. In particular, it would be nice to discuss what happens when the physics is a black box (i.e. we can interact with the system by applying control and observing, but we don't know the rules governing the physical system). Is this exactly when we are restricted to the "supervised" loss? Is there some middle ground? What if we had black box access to the exact physics, along with an approximate differentiable solver? This seems like a realistic scenario for e.g. large fluid flow scenarios.

**Experience Assessment:**

I have read many papers in this area.

**Review Assessment: Checking Correctness Of Derivations And Theory:**

I assessed the sensibility of the derivations and theory.

**Review Assessment: Checking Correctness Of Experiments:**

I assessed the sensibility of the experiments.

**Review Assessment: Thoroughness In Paper Reading:**

I read the paper at least twice and used my best judgement in assessing the paper.

---

> ### Author Response · Authors · 2019-11-12
> **Response to Review #3**
>
> Dear reviewer, thank you very much for your positive review of our manuscript and the helpful and constructive feedback. We have addressed your comments in a revised version of our paper, which we uploaded along with this comment. We have highlighted major changes in blue.
>
>
> > “Page 4: I found the statement "an agent trained in supervised fashion will then learn to average over the modes instead of picking one of them" a little confusing.”
> > “Page 5: I think the description of predictor-corrector could be clearer. In particular, I found the phrase "the correction uses o(t + ∆t) to obtain o(t + ∆t)" unclear.”
>
> We have revised the corresponding parts of our text to clarify these issues.
>
>
> > “Page 8: Could you add a description of what is observable to the body of the paper (I see it is included in the supplement)?”
>
> We added the explanation of observable quantities to the main paper.
>
>
> > “Page 8: I think ∇p needs to be divided by density in your NS equation”
>
> It is correct that the pressure gradient term is usually written with a 1/density term. As we are targeting incompressible flows, this corresponds to a global scaling factor, and in our simulations we normalize the fluid density to 1. This is clarified now in the appendix.
>
>
> > “Page 8 - 9: Is there a limit on the size of the force that can be applied at any point? I know the total force is penalized, but what about the maximum force applied at any point?”
>
> The force loss we chose sums the squares at each grid point. We now made this explicit in eq. 3 by adding the spatial integral. This loss formulation naturally penalizes force spikes and prefers forces spread out over space.
>
> We added histograms of force strengths for Burger’s and the direct control smoke example to the appendix. They show that the likelihood of large forces falls off approximately exponentially for our method. One could additionally enforce a limit on the force applied in a differentiable way, e.g. by using a sigmoid function. However, we did not find this necessary for our control experiments.
>
>
> > “I think "differentiable physics" losses need a more detailed explanation in the body of the paper.”
>
> We expanded the paragraph on differentiable physics losses in the Preliminaries section, giving a clear definition of what we mean by differentiable physics.
>
>
> > “In the supplement, it is defined using B_r, but I don't think B_r is defined.”
>
> B_r refers to a blur function. The blur helps make the gradients smoother and creates non-zero gradients in places where prediction and target do not overlap. We added the definition to the appendix.
>
>
> > “It seems like the differential physics loss requires a differential solver”
>
> Yes, we now clarify that in the Preliminaries section.
>
>
> > “what happens when the physics is a black box [...]? Is this exactly when we are restricted to the "supervised" loss?”
>
> If gradients for the physical behaviour are not available, reinforcement learning (RL) would be the method of choice. However, pure RL scales poorly with the number of controllable parameters, since an RL agent starts out by trying out random combinations of controls (see, e.g., https://fluxml.ai/2019/03/05/dp-vs-rl.html). Assuming there are m controls and n frames, an RL agent has to predict n*m values in order to get a single feedback.
>
> Our method could be combined with RL to make this task easier to learn. Consider the following training procedure:
> - Pretrain OPs and CFE using supervised losses
> - Refine CFE on 1-step sequences using RL. Here, only m controls need to be predicted (instead of n*m).
> - Refine OPs using RL, starting with short and progressing to longer sequences
> - Refine OP and CFE jointly with RL.
> Since each step builds upon the last, this should make it much easier for the model to learn sensible controls. This direction poses an interesting problem for future research.
>
>
> > “ Is there some middle ground? What if we had black box access to the exact physics, along with an approximate differentiable solver?”
>
> With approximate gradients, the CFE model could still be trained using the differentiable physics loss. Depending on the quality of the gradients, we might also be able to train some of the short-term OPs predictor networks using this loss. For longer sequences, the gradients will likely be unreliable so we can resort to either supervised training only, or utilize RL.

---

> > ### Comment · AnonReviewer3 · 2019-11-12
> > **Thanks for addressing my comments**
> >
> > I think these clarifications strengthen the paper and make it more accessible for the ICLR audience. My rating remains an 8; I think this paper would be a great addition to the conference.

---

### Official Review · AnonReviewer2 · 2019-10-27
**Official Blind Review #2**

**Rating:** 6

**Review:**

[Summary]

This paper proposes to combine deep learning and a differentiable PDE solver for understanding and controlling complex nonlinear physical systems over a long time horizon. The method introduces a predictor-corrector scheme, which employs a hierarchical structure that temporally divides the problem into more manageable subproblems, and uses models specialized in different time scales to solve the subproblems recursively.

For dividing the problem into subproblems, they use an observation predictor network to predict the optimal center point between two states. To scale the scheme to sequences of arbitrary length, the number of models scales with O(log N). For each subproblem, the authors propose to use a corrector network to estimate the control force to follow the planned trajectory as close as possible.

They have compared their method with several baselines and demonstrated that the proposed approach is both more effective and efficient in several challenging PDEs, including the incompressible Navier-Stokes equations.

[Major Comments]

Predicting the middle point between two states for modeling the dynamics via deep neural networks is not new, but I did not know any other works that use this idea for controlling PDEs.

I like the idea of splitting the control problem into a prediction and a correction phase, which leverages the power of deep neural networks and also incorporates our understanding of physics. The introduction of the hierarchical structure alleviates the problem of accumulating error in single-step forwarding models and significantly improves the efficiency of the proposed method. The videos for fluid control in the supplement materials also convincingly demonstrate the effectiveness of the technique.

I still have a few questions regarding the applicability and the presentation of the paper. Please see the following detailed comments.

[Detailed Comments]

In Section 3, the authors claim that their model "is conditioned only on these observables" and "does not have access to the full state." However, the model requires a differentiable PDE solver to provide the gradient of how interactions affect the outcome. These seem to contradict each other. Doesn't the solver require full-state information to predict the behavior of the system?

Related to the previous question, how can we make use of the differentiable PDE solver if we are uncertain or unknown of the underlying physics, i.e., partially observable scenarios.

The algorithm described in Section 5 seems to be the core contribution of this work. Instead of describing the algorithm in words, I think it would make it more clear if the authors can add an algorithm block in the main paper. It would also be better if the authors can include a few sentences describing the algorithm in the abstract to inform the readers of what to expect.

Figure 4 is a bit confusing, and it would be better if the authors can include the label for the x-axis. Besides, in the caption, the authors said that they show "the target state in blue." However, there are a lot of blue lines in the figure, and it is hard to know, at first glance, which one of them is the target.

In Table 1, the bottom two methods are using the same execution scheme and training loss, but the results are different. Is there a typo? Also, it would be better to bold the number that has the best performance.

**Experience Assessment:**

I have published one or two papers in this area.

**Review Assessment: Checking Correctness Of Derivations And Theory:**

I assessed the sensibility of the derivations and theory.

**Review Assessment: Checking Correctness Of Experiments:**

I assessed the sensibility of the experiments.

**Review Assessment: Thoroughness In Paper Reading:**

I read the paper at least twice and used my best judgement in assessing the paper.

---

> ### Author Response · Authors · 2019-11-12
> **Response to Review #2**
>
> Dear reviewer, thank you very much for your positive review of our manuscript and the helpful and constructive feedback. We have addressed your comments in a revised version of our paper, which we uploaded along with this comment. We have highlighted major changes in blue.
>
>
> > “Predicting the middle point [...] is not new, but I did not know any other works that use this idea for controlling PDEs.”
>
> The idea of splitting problems at the midpoint is prevalent throughout computer science. Popular examples are data structures like binary trees or sorting algorithms like quicksort. In the context of PDEs, higher-order solver schemes like Runge-Kutta predict the derivative at the midpoint in order to refine their solution. However, we are not aware of any other work that uses neural networks to predict midpoints of PDE sequences.
>
>
> > “the authors claim that their model [...] does not have access to the full state. Doesn't the solver require full-state information to predict the behavior of the system?”
>
> We apologize, this distinction was not properly explained in the paper. Both statements are correct: the differentiable solver has access to the full state during training and the machine learning models only see the observable values. Without the full state, the simulation cannot be performed properly. When deploying the trained agent to the real world, the simulation is replaced by real-world physics but the trained models can still infer control forces because they do not depend on the full state. I.e., our models only require the observations as inputs, not the full hidden state. We added this discussion to the Conclusions section.
>
>
> > “how can we make use of the differentiable PDE solver if we are uncertain or unknown of the underlying physics, i.e., partially observable scenarios.”
>
> In many cases, parts of the hidden information can be inferred from the observable values and the models will learn to do exactly that. However, to run the differentiable solver, full state information is required. Given a partially observed system that should be included in the training process, one could randomly generate a range of full states that project onto the observations and train on all of these.
>
>
> > “it would make it more clear if the authors can add an algorithm block in the main paper. It would also be better if the authors can include a few sentences describing the algorithm in the abstract to inform the readers of what to expect.”
>
> This is a good idea. We added the algorithm to the main text. In the abstract, we now outline how our method works and mention the two kinds of neural networks we use.
>
>
> > “Figure 4 is a bit confusing [...]”
>
> We now labelled the x-axis and plotted the initial and target state with dashed lines.
>
>
> > “In Table 1, the bottom two methods are using the same execution scheme and training loss, but the results are different.”
>
> Both rows refer to the same iterative algorithm, the difference being only the number of iterations performed. We added this information to the table.

---

> > ### Comment · AnonReviewer2 · 2019-11-15
> > **Response to Rebuttal**
> >
> > Thank you for your response, which addressed many of my concerns. I've read the updated version, and I think the paper looks much more clear with these included. I keep my rating as 6.

---

### Comment · Area_Chair1 · 2019-11-13
**Thanks for your reviews. Please take a look at the rebuttal.**

Dear reviewers,

Thank you very much for your efforts in reviewing this paper.

The authors have provided their rebuttal. It would be great if you take a look at them, and see whether it changes your opinion in anyway (some of you have already done this). If there is still any unclear point or a serious disagreement, please bring it up. Also if you are hoping to see a specific change or clarification in the paper before you update your score, please mention it.

The authors have only until November 15th to reply back.

I also encourage you to take a look at each others’ reviews. There might be a remark in other reviews that changes your opinion.

Thank you,
Area Chair

---

### Decision · Program_Chairs · 2019-12-19

**Decision:**

Accept (Spotlight)

**Comment:**

The paper proposes a method to control dynamical systems described by a partial differential equations (PDE). The method uses a hierarchical predictor-corrector scheme that divides the problem into smaller and simpler temporal subproblems. They illustrate the performance of their method on 1D Burger’s PDE and 2D incompressible flow.
The reviewers are all positive about this paper and find it well-written and potentially impactful. Hence, I recommend acceptance of this paper.